



# Performance of ensemble streamflow forecasts under varied hydrometeorological conditions

Harm-Jan F. Benninga[1,*], Martijn J. Booij[1], Renata J. Romanowicz[2], Tom H.M. Rientjes[3]

[1]Water Engineering and Management, Faculty of Engineering Technology, University of Twente, 7500 AE Enschede, The Netherlands
[2]Institute of Geophysics, Polish Academy of Sciences, 01-452 Warsaw, Poland
[3]Water Resources, Faculty of Geo-Information Science and Earth Observation, University of Twente, 7500 AE Enschede, The Netherlands
[*]Present address: Water Resources, Faculty of Geo-Information Science and Earth Observation, University of Twente

Correspondence to: Harm-Jan F. Benninga (h.f.benninga@utwente.nl)

**Abstract.** The paper presents a methodology to give insight in the performance of ensemble streamflow forecasting systems. We developed an ensemble forecasting system for the Biała Tarnowska, a mountainous river catchment in southern Poland, and analysed the performance for lead times from 1 day to 10 days for low, medium and high streamflow and related runoff generating processes. Precipitation and temperature forecasts from the European Centre for Medium-Range Weather Forecasts serve as input to a deterministic lumped hydrological (HBV) model. Due to inconsistent bias, the best streamflow forecasts were obtained without pre- and post-processing of the meteorological and streamflow forecasts. Best forecast skill, relative to alternative forecasts based on historical measurements of precipitation and temperature, is shown for high streamflow and for snow accumulation low streamflow events. Forecasts of medium streamflow events and low streamflow events generated by precipitation deficit show less skill. To improve the performance of the forecasting system for high streamflow events, in particular the meteorological forecasts require improvement. For low streamflow forecasts, the hydrological model should be improved. The study recommends improving the reliability of the ensemble streamflow forecasts by including the uncertainties in hydrological model parameters and initial conditions, and by improving the dispersion of the meteorological input forecasts.

## 1 Introduction

Accurate flood forecasting (Cloke and Pappenberger, 2009; Penning-Rowsell et al., 2000; Werner et al., 2005) and low streamflow forecasting (Demirel et al., 2013a; Fundel et al., 2013) are important to mitigate the negative effects of extreme events by enabling early warning. Accurate forecasting becomes increasingly more important since frequency and magnitude of low and high streamflow events are projected to increase in many areas in the world (IPCC, 2014). Due to socio-economic development also the impacts of extreme events increase (Bouwer et al., 2010; Rojas et al., 2013; Wheater and Gober, 2015).





Hydrological forecasting systems are often implemented as ensemble forecasting systems (Cloke and Pappenberger, 2009). Ensemble forecasts provide information about the possibility that an event occurs (Thielen et al., 2009), and allow quantification of the forecast uncertainty (Zappa et al., 2011). Uncertainties in streamflow forecasts originate from meteorological input, and hydrological model parameters, initial conditions and model structure (Cloke and Pappenberger, 2009; Demirel et al., 2013a; Zappa et al., 2011).

A number of studies investigated the performance of ensemble forecasting systems for different lead times, e.g. Ye et al. (2014) for the European Centre for Medium-Range Weather Forecasts (ECMWF) medium-range ensemble precipitation forecasts, Alfieri et al. (2014) for the European Flood Awareness System (EFAS), and Bennett et al. (2014), Olsson and Lindström (2008), Renner et al. (2009) and Roulin and Vannitsem (2005) for several catchments varying in size and other characteristics. They all found a deterioration of performance with increasing lead time. EFAS serves to provide high streamflow forecasts in large European river catchments for lead times between 3 and 10 days (Thielen et al., 2009). Relative to hydrological persistency the system skilfully forecasts high streamflow events for all lead times up to 10 days, with increasing skill for larger upstream areas (Alfieri et al., 2014). In EFAS critical flood warning thresholds are based on simulated streamflow, because model results and streamflow measurements can largely deviate (Thielen et al., 2009). EFAS is aimed at providing early warnings of possible flooding, instead of providing specific river streamflow forecasts (Demeritt et al., 2013). Most studies on medium-range ensemble streamflow forecasting focused either on flood forecasts (e.g. Alfieri et al., 2014; Bürger et al., 2009; Komma et al., 2007; Olsson and Lindström, 2008; Roulin and Vannitsem, 2005; Thielen et al., 2009; Zappa et al., 2011) or low streamflow forecasts (Demirel et al., 2013a; Fundel et al., 2013), in contrast to studies to general ensemble streamflow forecasting systems (Bennett et al., 2014; Demargne et al., 2010; Renner et al., 2009; Verkade et al., 2013). For two Belgium catchments the high streamflow forecasting system of Roulin and Vannitsem (2005) is more skilful for the winter period than the summer period. Previous studies did not assess effects of runoff processes, like snowmelt and extreme rainfall events, on the performance of the ensemble forecasts.

Information on the relative importance of uncertainty sources in forecasts is helpful to improve the forecasts effectively (Yossef et al., 2013). A number of studies report on how errors in the meteorological forecasts and the hydrological model contribute to errors in medium-range hydrological forecasts. Demargne et al. (2010) show that hydrological model uncertainties (initial conditions, model parameters and model structure) are most significant at short lead times. This also depends on the flow category. Hydrological model uncertainties significantly degrade the evaluation score up to a lead time of 7 days for all flows and up to a lead time of 2 days for the very high streamflow events. Renner et al. (2009) found an underprediction of low forecast probabilities (few ensemble members over a high streamflow threshold), which they attribute to the meteorological forecasts (insufficient variability). On the other hand the high forecast probabilities (low threshold) are overpredicted, which Renner et al. (2009) attribute to both the hydrological model and the meteorological input data. Olsson and Lindström (2008) found an underestimation of the spread of ensemble flood forecasts, to an extent that decreases with lead time. They conclude that the meteorological forecasts and the hydrological model have a comparable contribution to this underestimation. In addition Olsson and Lindström (2008) show overprediction of forecast



probabilities over high thresholds, which they mainly attribute to the meteorological forecasts. Regarding low streamflow forecasts, Demirel et al. (2013a) concluded that uncertainty of hydrological model parameters has the largest effect, whereas meteorological input uncertainty has the smallest effect. Based on those studies we can say that for high streamflow forecasts uncertainties in the meteorological forecasts are dominant, whereas for low streamflow the uncertainties in the hydrological model become more important.

The objective of this study is to investigate the performance and limitations of ECMWF based ensemble streamflow forecasting for lead times up to 10 days for low, medium and high streamflow in a catchment with seasonal variation in runoff generating processes. We aim to evaluate whether performance of the forecasting system can be related to specific runoff generating processes based on hydrometeorological conditions. Further, we assess whether the main source of forecast error relates to the meteorological input or to deficiencies of the hydrological model for the different streamflow categories and runoff generating processes.

## 2 Study catchment and data

### 2.1 Study area and measurement data

The Biała Tarnowska catchment in Poland serves as study area. This catchment is selected because of its large variation in streamflow, with seasonal variation in runoff generating processes. The catchment (Fig. 1) is located in a mountainous part of southern Poland. Napiorkowski et al. (2014) further describe the catchment. The River Biała Tarnowska discharges into the River Dunajec, which is a tributary of the River Vistula. The length of the river is 101.8 km with catchment area 956.9 km$^2$. The mean streamflow discharge (1972–2013) is 9.4 m$^3$ s$^{-1}$. Streamflow is characterized by large variation and extreme high flows with highest measured streamflow of 611 m$^3$ s$^{-1}$. During winter and spring snowmelt plays an important role. Comparison of the time series of precipitation and streamflow reveals that the lag time between intense precipitation events and related peaks in streamflow varies between 1 and 3 days.

Precipitation, temperature and streamflow measurement series are available at a daily time interval for the period 1 January 1971 to 31 October 2013, provided by the Polish Institute of Meteorology and Water Management. Precipitation and temperature data from 5 measurement stations (Fig. 1) have been selected because of their distribution over the catchment and data series completeness. The data are spatially interpolated based on Thiessen polygons (Fig. 1) to represent catchment averages. Given that stations are mostly located in valleys and precipitation and temperature vary with elevation, the catchment averages are biased (Panagoulia, 1995; Sevruk, 1997). Following Akhtar et al. (2009), precipitation measurements are corrected using relative correction factors (in %) whereas temperature measurements are corrected using absolute correction factors (in °C). The precipitation correction factor differs considerably between months. For December–February the mean precipitation gradient is 10.5 % 100 m$^{-1}$, while for March–November the mean precipitation gradient is 5.4 % 100 m$^{-1}$. Although the number of stations is limited to accurately determine precipitation and temperature gradients, the calculated precipitation gradients are used because of the clear difference between two periods. The temperature gradient





does not vary much over the year and therefore the global standard temperature lapse rate of 0.65 °C 100 m$^{-1}$ is applied. By the corrections the annual mean precipitation increases from 741.2 mm to 768.4 mm and the annual mean potential evapotranspiration decreases from 695.3 mm to 674.4 mm.

## 2.2 Meteorological forecast data

The meteorological ensemble forecast data from ECMWF are used, because of the good performance compared to other meteorological ensemble forecast sets (Buizza et al., 2005; Tao et al., 2014) and because these forecasts are frequently used in hydrological ensemble forecasting (Cloke and Pappenberger, 2009). Persson and Andersson (2013) and ECMWF (2012) describe how ECMWF generates the meteorological ensemble forecasts. The ensemble forecasts consist of one control forecast (no perturbation) and 50 ensemble members. The ensemble members should represent initial condition and
meteorological model uncertainty (Leutbecher and Palmer, 2008; Persson and Andersson, 2013).

      The THORPEX Interactive Grand Global Ensemble (TIGGE) project, developed by The Observing System Research and Predictability Experiment (THORPEX), provides historical forecast data from 1 October 2006 onwards (Bougeault et al., 2010). The resolution of the ensemble and control forecasts is 32 km × 32 km (ECMWF, 2012). Using the TIGGE data portal we interpolated the forecasts to a regular grid (Bougeault et al., 2010) with a resolution of 0.25° × 0.25°
(~17.9 km × 27.8 km at this latitude). In this study a maximum lead time of 10 days is used, following the World Meteorological Organization (WMO) that defines medium-range as forecasts with lead times from 3 days to 10 days (ECMWF, 2012). We also refer to Alfieri et al. (2014), Bennett et al. (2014), Demirel et al. (2013a), Olsson and Lindström (2008), Renner et al. (2009), Roulin and Vannitsem (2005) and Verkade et al. (2013) that use 9 or 10 days as maximum lead time. Because we use a lumped hydrological model with a daily time step (Sect. 3.1.1), we averaged daily ECMWF forecasts
according to the relative area coverage of the seven grid cells that overlay the catchment.

      According to Persson and Andersson (2013) ECMWF forecasts may apply to a land elevation that significantly differs from the actual elevation in a grid and this can lead to biases. In this study correction for such elevation errors is ignored since any systematic bias is accounted for in the pre-processing step (Sect. 3.1.3). ECMWF provides temperature forecasts at 00:00 hr. or 12:00 hr. This means that temperature forecasts cannot be considered as representative for one day.
To obtain representative daily average temperature forecasts we weight the temperature forecasts at 00:00 hr., 12:00 hr and 24:00 hr by 25%, 50% and 25% respectively.

## 3 Methodology

## 3.1 The ensemble streamflow forecasting system

The ensemble streamflow forecasting system consists of multiple components, presented in Fig. 2. Uncertainties in
meteorological forecasts, model parameters, model initial conditions and model structure affect ensemble streamflow forecasts (Cloke and Pappenberger, 2009; Demirel et al., 2013a; Zappa et al., 2011). Bennett et al. (2014) and Cloke and





Pappenberger (2009) describe that uncertainties in meteorological forecasts are the largest source of uncertainty beyond 2–3 days, and therefore only meteorological forecast uncertainty is incorporated in many studies (Bennett et al., 2014). By considering only uncertainty of the meteorological forecasts we focus on the effect of ensemble meteorological forecasts on streamflow forecasts.

### 3.1.1 Hydrological model

The hydrological model we use is a lumped Hydrologiska Byråns Vattenbalansavdelning (HBV) model that we run at daily time step by available hydrometeorological time series data for streamflow, gauged precipitation and temperature, and ECMWF meteorological forecasts. The model has 14 parameters and includes a snow accumulation and melting routine (Lindström et al., 1997; Osuch et al., 2015). Daily potential evapotranspiration rates are based on air temperature following the method of Hamon (Lu et al., 2005). The HBV model has wide application in studies on ensemble streamflow forecasting (e.g. Cloke & Pappenberger, 2009; Demirel et al., 2013a, 2015; Kiczko et al., 2015; Olsson & Lindström, 2008; Renner et al., 2009; Verkade et al., 2013). The choice for a lumped model with a daily time step is basically the result of the spatial and temporal resolution of the available data. The River Rhine forecasting suite also adopts the HBV model at a daily time step that is applied as semi-distributed model to 134 sub catchments (Renner et al., 2009). The catchment area of Biała Tarnowska ($\sim$1000 km$^2$) is comparable to the area of the sub catchments in the River Rhine forecasting suite.

The HBV model is calibrated using the differential evolution with global and local neighbourhoods (DEGL) method, described by Das et al. (2009). The settings that we used are adopted from the best performing variant of Das et al. (2009) (maximum number of model runs is 50000). The model is calibrated over the period 1 November 1971 to 31 October 2000 with the time series of precipitation and temperature as input and streamflow measurements as reference output. The validation period is 1 November 2000 to 31 October 2013. The objective function selected for calibration is $Y$, which combines the Nash–Sutcliffe coefficient (NS) and the relative volume error ($E_{RV}$) (Akhtar et al., 2009; Rientjes et al., 2013). According to Rientjes et al. (2013) values of $Y$ below 0.6 indicate poor to satisfactory performance. The model parameters were drawn uniformly from predefined parameter ranges (Osuch et al., 2015).

### 3.1.2 Updating of initial states

Hydrological forecasting often relies on the updating of hydrological model storages to best represent the hydrological conditions in the catchment at the forecast day (e.g. Demirel et al., 2013a; Werner et al., 2005; Wöhling et al., 2006). For storage updating we follow Demirel et al. (2013a) and apply a procedure based on measured streamflow on the day preceding the forecast day. The measured streamflow of the day preceding the forecast day is divided in a fast and a slow runoff component to update the fast runoff reservoir and the slow runoff reservoir in the HBV model. To determine the ratio between the fast and slow components a relationship between total simulated streamflow and the fraction of fast runoff is established. However, this relationship contains large uncertainty. For example, for a total simulated streamflow of 10 m$^3$ s$^{-1}$ the fraction varies between 0 and 0.6 and for a streamflow of 20 m$^3$ s$^{-1}$ it varies between 0.3 and 0.7. To reduce uncertainty





in the fraction of fast runoff the storage of the fast runoff HBV reservoir and net inflow in the fast runoff reservoir are both tested as an additional descriptor of the relationship between streamflow and the fraction of fast runoff.

### 3.1.3 Pre- and post-processing

Errors in meteorological forecasts as well as in hydrological models introduce biases (Cloke and Pappenberger, 2009;
Verkade et al., 2013). Several studies suggest that post-processing of streamflow forecasts is more effective to improve the forecast skill than pre-processing of meteorological input data (Kang et al., 2010; Verkade et al., 2013; Zalachori et al., 2012). Verkade et al. (2013) and Zalachori et al. (2012) found that corrections made to meteorological forecasts lose their effect when propagated through a hydrological model (Verkade et al., 2013; Zalachori et al., 2012). Results by Zalachori et al. (2012) indicate that combined pre- and post-processing results in best forecast quality. In this study both pre-processing
of the meteorological input forecasts and post-processing of the streamflow forecasts are tested.

Many studies used (conditional) quantile mapping (QM) for pre-processing (Boé et al., 2007; Déqué, 2007; Kang et al., 2010; Kiczko et al., 2015; Verkade et al., 2013; Wetterhall et al., 2012) and post-processing (Hashino et al., 2007; Kang et al., 2010; Madadgar et al., 2014; Shi et al., 2008) to correct for bias and dispersion errors. According to Kang et al. (2010) QM generally performs well in both pre- and post-processing. Hashino et al. (2007) advise to use QM, because of the good
performance regarding sharpness and discrimination and the simplicity of the method. The principle of QM is that the cumulative distribution function (CDF) of the forecasts over a control period is matched to the CDF of the measurements over the same period, after which a correction function is generated (Boé et al., 2007). This means that the correction is conditional on the value of the forecasted variable itself. Boé et al. (2007), Déqué (2007) and Madadgar et al. (2014) further explain QM. The empirical CDFs of the measurements and forecasts are established on the training period 1 November 2011
to 31 October 2013 (two hydrological years) and validated over the period 1 November 2007 to 31 October 2011.

Distributions can be different for different lead times and weather patterns or seasons (Boé et al., 2007; Wetterhall et al., 2012), so three QM set-ups are tested with or without distinguishing different lead times and seasons. Combining pre-processing and post-processing results in four processing strategies. In strategy 0 no pre- and post-processing are applied. In strategy 1 and 2 QM is applied to pre-process the meteorological forecasts, respectively without post-processing and with
post-processing. In strategy 2 the post-processing is performed based on the correction between 'perfect forecasts' (streamflow simulations with input from measurements) and streamflow measurements to account for hydrological model uncertainties (Verkade et al., 2013). In strategy 3 only post-processing is applied, based on the correction between streamflow forecasts generated with uncorrected meteorological forecasts and measured streamflow. In this strategy meteorological and hydrological model uncertainties are treated together (Verkade et al., 2013).

**3.2 Evaluation scores of the ensemble forecasts**

To measure general quality and skill of the streamflow forecasts, the continuous ranked probability score (CRPS) and the continuous ranked probability skill score (CRPSS) are used. According to Demargne et al. (2010) and Hamill et al. (2000) a



single evaluation score is inadequate to evaluate the overall performance of a forecasting system. Three properties of forecast quality are reliability, sharpness and resolution (WMO, 2015). Reliability refers to the statistical consistency between measurements and simulations (Candille & Talagrand, 2005; Velázquez et al., 2010) and whether uncertainty is correctly represented in the forecasts (Bennett et al., 2014). We evaluate reliability by rank histograms and reliability diagrams.

Sharpness is defined as the tendency to forecast probabilities of occurrence near 0 or 1, as opposed to values clustered around the mean (climatological) probability (Ranjan, 2009; WMO, 2015). If an ensemble forecasting system always forecasts an event probability close to climatological probability, instead of close to 0 or close to 1, this forecasting system is not useful, although it might be well calibrated (Ranjan, 2009; Wilks, 2006). The histograms accompanying reliability diagrams are used to evaluate sharpness. Resolution is the ability of the forecast model to correctly forecast the occurrence or

nonoccurrence of events (Demirel et al., 2013a; Martina et al., 2006). We employ relative operating characteristics (ROC) curves to evaluate resolution.

### 3.2.1 Continuous ranked probability score

The CRPS is an overall, single-number score for judging the quality of probabilistic forecasts (Hamill et al., 2000). CRPS measures the error of the ensemble forecasts by integrating the squared distance between the CDFs of the forecasts and a

reference streamflow (Bennett et al., 2014; Demargne et al., 2010; Verkade et al., 2013). The score is frequently used in atmospheric (Velázquez et al., 2010) and hydrological sciences (Bennett et al., 2014; Pappenberger et al., 2015; Velázquez et al., 2010) and in most cases it is the recommended evaluation score for ensemble forecasts (Pappenberger et al., 2015). CRPS is sensitive to the entire range of the variable of interest and it does not require the introduction of predefined classes (Hersbach, 2000). A CRPS of 0 indicates a perfect simulation, which can only be achieved in the case of a perfect

deterministic forecast (Hersbach, 2000). Because in practice CRPS approaches the average value of the evaluated variable (with the same unit), the score cannot directly be compared among different areas, seasons or streamflow categories (Ye et al., 2014). Comparison between different lead times is possible, as average streamflow values do not change with lead time.

### 3.2.2 Continuous ranked probability skill score

Normalizing the CRPS against the CRPS of alternative forecasts eliminates the effect of the magnitude of the investigated

variable and compares the forecasts with a relevant alternative forecast (i.e. skill), used by e.g. Bennett et al. (2014), Demargne et al. (2010), Renner et al. (2009), Velázquez et al. (2010) and Verkade et al. (2013). The CRPSS is defined as:

$$CRPSS = 1 - \frac{CRPS_{forecasts}}{CRPS_{alternative}}, \tag{1}$$

A system with perfect skill results in a CRPSS of 1 and a negative CRPSS indicates that the forecasting system performs worse than the alternative forecasts (Demargne et al., 2010; Ye et al., 2014). To evaluate skill of the forecasting system we

define the alternative forecast set as forecasts that are generated without using meteorological forecasts. It is common practice to apply hydrological persistency or hydrological climatology as alternative forecast set (Bennett et al., 2014).





However, Pappenberger et al. (2015) argue that this can result in an overestimation of forecast skill because other alternative forecast sets might be more difficult to beat in performance. Following Bennett et al. (2013), Bennett et al. (2014) and Pappenberger et al. (2015) the most appropriate alternative forecast set is selected based on their CRPS results. We use a single alternative forecast set for all streamflow categories, so one CRPS$_{alternative}$ is calculated. With hydrological persistency

the most recent streamflow measurement (i.e., from the day preceding the forecast day) serves as forecast for all lead times. Regarding hydrological climatology, the average measured streamflow, after a smoothing window of 31 days, on the same calendar day over the last 20 years is used, following Bennett et al. (2013). For streamflow forecasts based on an ensemble of historic measurements of precipitation and temperature, measurements on the same calendar day over the past 20 years are used, after Pappenberger et al. (2015).

10          The results in Fig. 3 indicate that forecasts based on meteorological climatology result in the best CRPS scores and thus imply to be the most appropriate alternative streamflow forecasts, as also found in the studies of the Bennett et al. (2013), Bennett et al. (2014) and Pappenberger et al. (2015).

### 3.2.3 Rank histogram

Rank histograms enable to diagnose average errors in the mean and spread (under- or overdispersion) of the ensemble

forecasts (Hamill, 2001; Hamill et al., 2000) and according to Wilks (2006) they are commonly used to evaluate the reliability (or consistency) of ensemble forecasts. The consistency condition states that the reference streamflow is just one more member of the ensemble and it should be statistically indistinguishable from the ensemble forecast (Wilks, 2006). To construct a rank histogram, the reference streamflow is added to the ensemble forecast set and the histogram is constructed from the ranks of the reference streamflow (Velázquez et al., 2010). In an ensemble forecasting system with perfect spread

each ensemble member is equally likely, so all reference streamflow ranks are equally likely and the rank histogram is uniform (Hamill, 2001; Hersbach, 2000; Wilks, 2006; WMO, 2015; Zalachori et al., 2012).

         To indicate the flatness of rank histograms Candille and Talagrand (2005) propose a numerical indicator $\delta$. Because $\delta$ is proportional to the length of the time series (Velázquez et al., 2010), we use the Mean Absolute Error as flatness coefficient $\varepsilon$:

$\varepsilon = \frac{1}{n+1} \sum_{z=1}^{z=n+1} |f(z) - y|$ ,                                                                                                 (2)

$f(z)$ = Relative frequency of reference streamflow in rank $z$ [-]

$y = \frac{1}{n+1}$ = Theoretical relative frequency (uniform distribution) [-]

$n$ = Number of ensemble members [-]

In a perfectly consistent forecasting system the relative frequency in each rank is equal to the relative frequency according to

uniform distribution. This gives an optimum value of $\varepsilon$ equal to 0. The rank histogram and flatness coefficient contain a random element if multiple ensemble members and the measurement have the same value, like 0 mm precipitation (Hamill and Colucci, 1998).



### 3.2.4 Reliability diagram

The reliability diagram is a common way to summarize and evaluate reliability of probabilistic forecasting systems (Bröcker and Smith, 2007). The diagram plots observed relative frequency against the predicted probability for a certain event (Bröcker and Smith, 2007; Demirel et al., 2013a). For a well calibrated forecasting system the reliability diagram is close to

the 1:1 diagonal (Ranjan, 2009; WMO, 2015). The five forecast probability bins that we use are 0%–20%, 20%–40%, … and 80%–100%, which were also used by Demirel et al. (2013a) and Bennett et al. (2014). Following Bröcker and Smith (2007) the observed frequencies are plotted against the average of forecast probabilities per bin instead of the bin centre %. Plotting against bin centres (so 10%, 30%, etc.) can cause substantial deviations from the diagonal.

The histogram showing sample size in each probability bin indicates the sharpness of forecasts (Ranjan, 2009;

Renner et al., 2009; WMO, 2015).

### 3.2.5 Relative operating characteristic

Contingency tables and ROC curves analyze whether the forecast model correctly forecasts the occurrence and nonoccurrence of events. To establish the ROC a set of contingency tables is made, one for each examined probability threshold and these form a hit rate/false alarm rate graph for one predefined flow threshold (Atger, 2001; Buizza et al., 1999;

Fawcett, 2006; WMO, 2015). The area under the ROC curve (AUC) can be used to obtain a single score for performance (Fawcett, 2006; Wilks, 2006). A perfect ensemble forecasting system has an area of 1 under the ROC curve (100% hit rate, 0% false alarm rate for all probability thresholds), while a forecasting system with zero skill has a diagonal ROC curve with an area of 0.5 (coincides with diagonal) (Fawcett, 2006; Velázquez et al., 2010; WMO, 2015). Buizza et al. (1999) state that it is common practice to consider an area of more than 0.7 as indicative for useful prediction systems and 0.8 for good

prediction systems.

### 3.3 Investigation of error contributors

The evaluation of ensemble streamflow forecasts is affected by errors from the meteorological forecasts, the hydrological model (including errors in the initial conditions) and errors in the measurements that serve as reference streamflow (Renner et al., 2009). By evaluation against perfect forecasts the streamflow measurement error and the hydrological model error are

eliminated, because both the ensemble streamflow forecasts and the reference streamflow contain these errors (Demargne et al., 2010; Olsson and Lindström, 2008; Renner et al., 2009). If we neglect measurement errors, evaluation against streamflow measurements ($CRPS_{meas}$) contains errors from the meteorological forecasts and the hydrological model and evaluation against perfect streamflow forecasts ($CRPS_{sim}$) exclusively contains errors from the meteorological forecasts (Demargne et al., 2010; Olsson and Lindström, 2008; Renner et al., 2009). A low $CRPS_{sim} / CRPS_{meas}$ ratio means that the

hydrological model errors are dominant and a high ratio means that the meteorological errors are dominant.





## 3.4 Evaluation of streamflow categories

We evaluate the forecasting system for different streamflow categories as defined in Table 1. A low streamflow threshold $Q_{75}$ (exceedance probability of 75%) guarantees that a sufficient number of events are considered in the evaluation of this streamflow category while streamflow below this threshold still affects river functions (Demirel et al., 2013b). Similarly, we used $Q_{25}$ as high streamflow threshold.

## 3.5 Evaluation of runoff generating processes

The high streamflow forecasts and low streamflow forecasts are evaluated for the various hydrometeorological conditions that can generate these events. Medium flows are not evaluated for different runoff generating processes since these events commonly result from a combination of runoff generating processes under non-extreme hydrometeorological conditions.

### 3.5.1 High streamflow generating processes

Various runoff contributing processes can result in high flows. Table 2 defines the processes and classification rules we use in this study, based on the processes Merz and Blöschl (2003) distinguish. The classification rules are based on fluxes and storages at one day before the event, because in the HBV model it takes one modelling time step before the rainfall and snowmelt fluxes end up in the fast runoff and slow runoff reservoirs and can form runoff.

Figure 4a presents the distribution of high streamflow generating processes following the classification rules in Table 2. The figure shows an expected distribution of processes for this region.

### 3.5.2 Low streamflow generating processes

Processes that result in low flows are snow accumulation and the combination of low rainfall and high evapotranspiration over a period (precipitation deficit). Table 3 further characterizes these processes.

Figure 4b shows that these classification rules result in a reliable distribution of low streamflow generating processes over the year for this region.

## 4 Results

## 4.1 Ensemble streamflow forecasting system

### 4.1.1 Calibration and validation of the hydrological model

In Table 4 calibration and validation results are presented. The hydrological model performs better with corrected input data as compared to uncorrected input data. This implies that the systematic underestimation of precipitation and systematic overestimation of temperature (Sect. 2.1) are not fully captured in the calibration.



The updating of initial states of the fast runoff reservoir and slow runoff reservoir (Sect. 3.1.2) results in an improvement of $Y$ from 0.72 to 0.81 over the validation period. This effect decreases with lead time, but it is still noticeable at a lead time of 10 days ($Y = 0.730$ against $Y = 0.718$). Relating the fraction of fast runoff additionally to the storage of the fast runoff reservoir storage or net inflow does not result in a significant improvement of $Y$ compared to the original updating model. Therefore the original updating model, introduced by Demirel et al. (2013a), is used.

Simultaneous measurements and ECMWF forecasts are available over the period 1 November 2006 to 31 October 2013. In the hydrological year 2007 (1 November 2006 to 31 October 2007) the agreement between streamflow measurements and simulations is poor. Also with another model (data based mechanistic methodology (DBM)), with the same measurement data the performance was worse during this year (Kiczko et al., 2015). This is the result of measurement errors and/or human influence, because it is unlikely that in this period different hydrological processes are taking place that are not captured well by the HBV model and the DBM model. Therefore the period 1 November 2006 to 31 October 2007 is excluded from the evaluation period.

Table 5 presents the performance of the hydrological model for different lead times and streamflow categories. The NS values for the low and medium streamflow categories are negative, meaning that the averages of streamflow measurements in these categories are a better approximation of the measurements than the simulations. All measures highlight that the calibration is skewed to high streamflow situations, which is the result of the selected objective function that includes NS (Gupta et al., 2009). Gupta et al. (2009) also found that model calibration with NS tends to underestimate the low and high streamflow peaks.

The results in Table 5 improve considerably as a result of the updating of initial storages, especially for the low streamflow simulations. The effectiveness of the updating procedure depends on the autocorrelation of daily streamflow, because the updating is based on streamflow measurements of the preceding day. In low streamflow periods there is usually a high autocorrelation of daily streamflow, in contrast to high streamflow periods.

### 4.1.2 Pre- and post-processing strategy results

The best precipitation forecasts are obtained when QM is applied separately to each lead time, whereas the best temperature forecasts are obtained if, in addition, separate relationships for the summer and winter season are applied. The CRPS and Relative Mean Absolute Error ($E_{RMA}$) of the precipitation and temperature forecasts improve slightly and the flatness coefficients improve considerably as a result of the pre-processing.

Regarding the combined pre- and post-processing strategies, the results (not shown in the paper) indicate that strategy 0 results in the best CRPS and flatness coefficients of streamflow simulations.



### 4.2 Forecast performance

#### 4.2.1 Forecast skill

The streamflow forecasts are evaluated over the period 1 November 2007 to 31 October 2013 for lead times from 1 day to 10 days and for the different streamflow categories (defined in Table 1). The results are presented in Fig. 5. The CRPS increases

with lead time for all streamflow categories (Fig. 5a), so the performance of the streamflow forecasts deteriorates with lead time. For all streamflow together the CRPSS is positive for all lead times (Fig. 5b), so on average the streamflow forecasts are better than the alternative forecasts. This forecast skill is generated by the ECMWF forecasts compared to historical meteorological measurements.

Fig. 5b shows that the forecast skill is very different for the low, medium and high streamflow forecasts. The low

skill of low streamflow forecasts, especially for small lead times, can be explained by the important role of the initial conditions in the hydrological model. In low streamflow situations runoff is mainly generated by available resources in the catchment instead of precipitation input. Since the same initial model conditions are used to simulate the alternative forecasts, it is difficult to generate skilful low streamflow forecasts for small lead times (<3 days). Also the origin of the alternative forecasts plays a role. Since low streamflow events normally occur in the same period of the year due to climatic

seasonality, it can be expected that historical measurements of precipitation and temperature on the same calendar day provide functional input. After all, the performance of the meteorological forecasts preceding these events contributes to the low skill. The negative skill at small lead times indicates that historical measurements of precipitation and temperature are even better forecasts than the meteorological ensemble forecasts from ECMWF for this category of flows. From a lead time of 3 days the accumulated effects of the meteorological forecasts are more skilful than historical meteorological

measurements.

The medium streamflow forecasts do not have clear positive skill for all lead times. This can be explained by the fact that historical streamflow measurements are most often around the medium streamflow, so forecasts based on historical measurements of precipitation and temperature will be a good approximation for these flows.

The system has a high positive skill in forecasting high streamflow. In general initial conditions are relatively less important

in these events, because of the amount of water usually added to the system. However, we note that this depends on the responsible runoff generating process (see results in Sect. 4.4.1). As a result the streamflow forecasts and reference forecasts can easier deviate. In addition, these events are less well captured in historical measurements and thus in the alternative forecasts. This is because high streamflow periods are in general less predictable by historical measurements, in particular in small catchments.

#### 4.2.2 Forecast quality

Fig. 6 presents the flatness coefficients. The high values indicate that the rank histograms are far from flat, especially for small lead times and low streamflow events. The rank histograms (not shown in the paper) are U-shaped, which indicates



underdispersion and/or conditional bias in the streamflow forecasts (Hamill, 2001). The rank histograms of the meteorological forecasts show that the ECMWF forecasts are also underdispersed, so this is one cause why the streamflow forecasts are underdispersed. In Sect. 5 the consequences of neglecting uncertainties in the hydrological model and initial conditions are further discussed.

5       The rank histograms of the different streamflow categories show that the streamflow forecasts contain a conditional bias. In general, high streamflow is underestimated by the forecasting system and this underestimation increases with lead time. On the other hand, low streamflow is generally overestimated. This can be the result of too coarse spatial and temporal model resolution. Using a lumped model and aggregating the meteorological input over the catchment flattens the extreme flow events.

Also the reliability diagrams indicate low reliability of the streamflow forecasts, especially for small lead times. It appears that for low streamflow forecasts the observed relative frequencies are underestimated. Regarding the high streamflow forecasts the observed relative frequencies are overestimated, although the rank histograms indicate that high streamflow is underestimated. This is possible because in a rank histogram the measurements and forecasts are compared directly, whereas in a reliability diagram the measurements and forecasts are compared to a streamflow threshold.

Histograms showing the sample size in each probability bin of the reliability diagrams indicate that the sharpness of the forecasts is good, because forecast probabilities of low and high streamflow are most often close to 0 or 1, instead of forecast probabilities close to the mean probability. The sharpness decreases with lead time.

      All AUC values are above 0.85, whereas Buizza et al. (1999) consider 0.8 as indicative for good prediction systems.

### 4.3 Dominant error contributors

Fig. 7 shows that the relative contribution of meteorological forecast errors increases and the relative contribution of hydrological model errors decreases with lead time, although the performance of the hydrological model also deteriorates with lead time (see Table 5). Two effects contribute to this. In the first place the meteorological forecasts get worse with lead time and the meteorological forecasts accumulate in the hydrological forecasting system with lead time. In the second place the effect of the initial conditions in the hydrological model at the forecast day becomes smaller at larger lead times, because

more water is added to the system.

      In high streamflow forecasts the contribution of meteorological forecast errors is relatively more important, while in low streamflow forecasts the contribution of hydrological model errors is relatively more important. Initial conditions have relatively less influence on high streamflow (discussed in Sect. 4.2.1). In addition the hydrological model performs better for high streamflow than for low streamflow situations (Table 5), so meteorological forecast errors are relatively more important

in high streamflow situations.



### 4.4 Forecast skill for the runoff generating processes

#### 4.4.1 High streamflow generating processes

The highest skill is obtained for short-rain floods (Fig. 8a), at small lead times. Two effects explain this observation. First, long-rain floods and snowmelt floods are essentially driven by the water storage conditions in the catchment whereas in short-rain floods meteorological input has more influence. Figure 8b confirms the relative importance of meteorological forecasts in these events. This results in higher potential to generate forecast skill, already at small lead times. At larger lead times the accumulation of rainfall in the forecasting system becomes important, which is confirmed by the increasing contribution of meteorological forecast errors in long-rain floods and snowmelt floods. Long-rain floods are skilfully forecasted from a lead time of 3 days and snowmelt floods are skilfully forecasted from a lead time of 2 days.

Second, the short and heavy rain events preceding short-rain floods are less well captured in historical meteorological measurements than the longer term processes underlying long-rain floods and snowmelt floods. The below 0 skill of long-rain and snowmelt flood forecasts indicate that the meteorological forecasts at small lead times do not result in positive skill as compared to forecasts based on historical meteorological measurements. The forecast skills of short-rain floods and snowmelt floods decrease again at larger lead times. This is the result of a decreased performance of the meteorological forecasts preceding these events. The skill of short-rain flood forecasts decreases the most and at the shortest lead time.

#### 4.4.2 Low streamflow generating processes

Figure 9a shows that the low forecast skill of low streamflow is caused by the precipitation deficit process, whereas the forecast skill of low streamflow events that are generated by snow accumulation is rather high. The low forecast skill of the precipitation deficit generated low streamflow events can be explained by the fact that low rainfall periods often occur in the same period of the year, due to climatic seasonality, and are therefore well captured by historical meteorological measurements. Also the performance of meteorological forecast models may play a role. Meteorological models tend to forecast drizzle instead of zero precipitation (Boé et al., 2007; Piani et al., 2010) and pre-processing has not been applied to correct for this. The skill increases for larger lead times, so ECMWF meteorological forecasts accumulated in the forecasting system are better model inputs than historical measurements for larger lead times. The fact that the contribution of initial conditions at the forecast day decreases for larger lead times (also see Fig. 9b) adds to this skill.

The forecast skill for both snowmelt floods and snow accumulation generated low streamflow events decreases from a lead time of 8 days, which indicates a decreasing skill of ECMWF temperature forecasts for large lead times.

For low streamflow generated by snow accumulation and precipitation deficits, errors from the HBV model and initial conditions make up a large part of the total error (Fig. 9b).





## 5 Discussion

The developed methodology of analysing an ensemble streamflow forecasting system has been applied to the Biała Tarnowska catchment for a 6 year period. By this, findings by this study do not allow direct generalisation but serve ongoing discussions on improving streamflow forecasting. Also, a longer evaluation period would allow evaluation of more extreme definitions of high and low streamflow.

The best streamflow forecasts are obtained without pre- and post-processing. The effectiveness of QM depends on whether during the validation period the same bias is present between the CDF of the measurements and the CDF of the forecasts as during the training period. Figure 10 shows large differences in biases between different years and between the training period and the validation period, suggesting that bias is affected by randomness. The relatively short time series of forecasts constrains pre- and post-processing procedures, because different weather patterns cannot be well identified and with a longer period a more consistent bias distribution could be obtained. Limitations of QM, as described by Boé et al. (2007), might also play a role in the ineffectiveness of pre- and post-processing. In spite of the limitations of QM, over the training period the pre- and post-processing strategies result in an improvement of the evaluation scores (strategy 3 with seasonal distinction gives the best performance), which indicates the potential of processing with QM if a consistent bias is present. A problem in pre- and post-processing in general is that the joint distribution of measurements and forecasts is often nonhomogeneous in time by, for example, an improvement of forecasting systems over time (Verkade et al., 2013).

Uncertainties in the hydrological model and model initial conditions have been ignored in the forecasting system. Considering the rank histogram results this may have affected the streamflow forecasts of short lead times and low streamflow in particular. Regarding the main effect on short lead times Bennett et al. (2014) and Pagano et al. (2013) discuss similar findings. The lower flatness coefficients of high streamflow forecasts compared to low streamflow forecasts reflect that for high streamflow forecasts meteorological input is relatively more important.

The classification of low and high streamflow generating processes is based on information that is available from the HBV model and measurement data series. This provides more insight in the performance of the forecasting system than a seasonal characterisation. Some assumptions must be kept in mind when interpreting the results. It is assumed that snow accumulation before an event is embedded in the snowpack storage of the HBV model. If a snowpack is present the event is classified as snowmelt flood or snow accumulation low streamflow. The lumped model causes a simplification here, because when there is a snowpack present in the model there is not necessarily a snowpack that covers the whole catchment. If no snowpack is present, it is assumed that the low streamflow event or high streamflow event is caused by low or high rainfall. The threshold of 10 mm day$^{-1}$ (see Table 2) is an unrefined simplification to distinguish between short-rain floods and long-rain floods. The simple character of the classification rules especially has consequences for the classification of events that are caused by a combination of processes, which often occur in practice and result in the highest floods. Another point is that only short-term information (from the day preceding the forecast day) is used to classify the processes. The lag time between precipitation peaks and streamflow peaks does not necessarily match with the HBV model calculation time step and the



classification rules use. Consequently, a streamflow at the day following a high rainfall event is classified as a short-rain flood, whereas the real streamflow peak might come one day later.

In the hydrological model the lag time between a rainfall event and the streamflow peak is set to 1 day. However, the timing of a rainfall event during the day is very important, especially in a small catchment. Evaluation of forecast performance in this paper indicates that the lag time is critical in the forecasting system, especially for short-rain floods. The results in Fig. 8b show that the ratio between the CRPS against perfect forecasts and the CRPS against streamflow measurements is above 100% for short-rain floods. This means that these forecasts are closer to the measurements than to the perfect forecasts. The precipitation peak in the measurements and the precipitation peak in the meteorological forecasts can be shifted one day with respect to each other and this can cause that the timing of the peak of the streamflow forecasts better corresponds to the streamflow measurements than to the peak of the perfect streamflow forecasts.

It is not trivial to compare the CRPS results to results in other studies, because the value depends on the magnitude of the evaluated variable. A similarity between the results in this study and previous studies is that performance of the streamflow forecasts decreases with lead time. Since Bennett et al. (2014) use the same alternative forecast set, the CRPSS results can be compared. Although Bennett et al. (2014) use a very different forecasting system and apply it to different situations, the forecast skills are comparable to the forecast skills obtained in this study.

## 6 Conclusions

We developed a methodology to analyse an ensemble streamflow forecasting system. For the case study of the Biała Tarnowska catchment we conclude:

- There are large differences in forecast skill for different runoff generating processes, compared to alternative forecasts based on historical measurements of precipitation and temperature. The system skilfully forecasts high streamflow events, although the skill depends on the runoff generating process and lead time. Also low streamflow events that are generated by snow accumulation are skilfully forecasted. Since the hit rates are high compared to the false alarm rates, the system has potential to generate forecasts for these streamflow categories. Sharpness of the forecasts is good, although it decreases with lead time. Medium streamflow events and low streamflow events that are generated by a precipitation deficit are not skilfully forecasted.

- When this or any other forecasting system is (further) developed with the objective to generate more accurate high streamflow forecasts, it is recommended to focus on improving the meteorological forecast input because errors from the meteorological forecasts are dominant in high streamflow forecasts. This can be achieved by improving the meteorological forecasts (e.g. using the higher resolution forecasts from COSMO-LEPS (Renner et al., 2009)) or by improving the pre-processing step. To improve low streamflow forecasts it is recommended to focus first on the hydrological model performance. In this study the calibration of the hydrological model is skewed to high streamflow situations. An easy improvement of the forecasts can be achieved by calibrating the hydrological model



specifically on low streamflow events. Besides improvement of the hydrological model, further research should be done to improve the meteorological forecasts, especially the precipitation forecasts (problem of forecasting of drizzle). When the forecasting system is applied exclusively on low or high streamflow forecasts the alternative forecast set should be reconsidered.

5     •   To improve the reliability of the ensemble streamflow forecasts it is recommended to include uncertainties in hydrological model parameters and initial conditions. Particularly for low streamflow forecasts this is essential. The uncertainty in the relationship between the fraction of fast runoff and total streamflow to update initial states might be utilized to incorporate initial condition uncertainty. Since the precipitation and temperature forecasts are also underdispersed, we recommend to investigate how the reliability of the precipitation and temperature forecasts can 10     be improved, by adding meteorological forecasts from other forecasting systems ('super-ensembles') (Bennett et al., 2014; Bougeault et al., 2010; Fleming et al., 2015; He et al., 2009) or by improved pre-processing.

    •   Pre- and post-processing with QM was not effective. In the discussion several limitations of QM have been described. With a longer time series of forecasts and other (more sophisticated) techniques the meteorological and hydrological forecasts could potentially be improved.

15     •   It is recommended to extend the study to other catchments and (if possible) with longer forecast datasets, to investigate the generality of the results and to test more extreme high and low streamflow thresholds.

The findings only apply to the study catchment and the developed system set-up, but the presented methodology of analysing an ensemble streamflow forecasting system is generally applicable. The methodology provides valuable information about the forecasting system, in which situations it can be used, and how the system can be improved effectively.

20 **Acknowledgements**

Marzena Osuch (Institute of Geophysics Polish Academy of Sciences) and Adam Kiczko (Warsaw University of Life Sciences) are thanked for valuable discussions and support on methods.

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



**Figures**

**Figure 1: Location and overview of the Biała Tarnowska catchment**





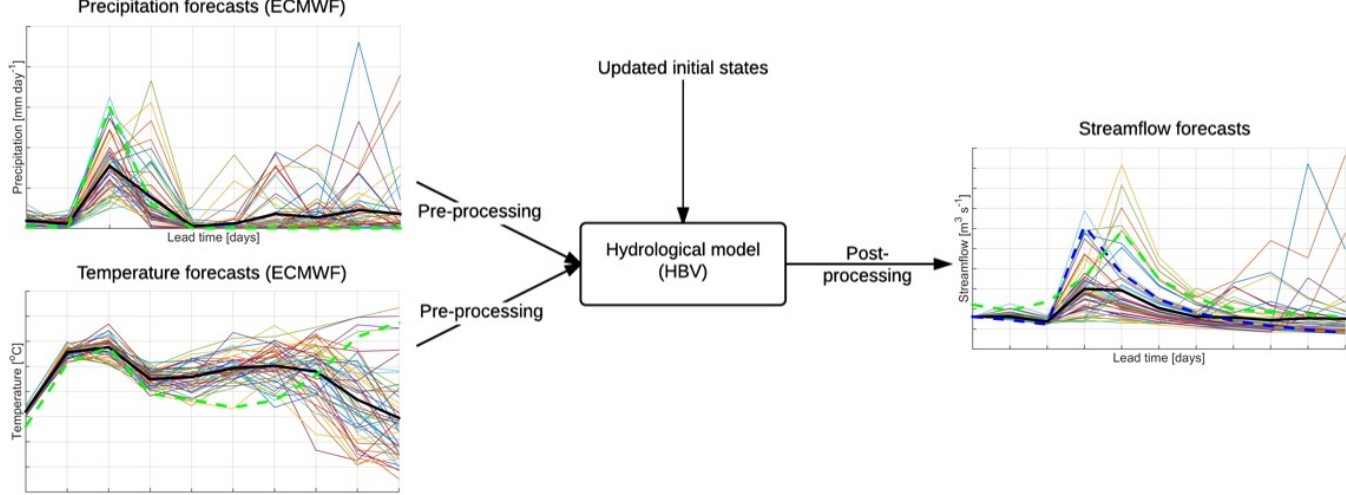

**Figure 2: Structure of the ensemble streamflow forecasting system**

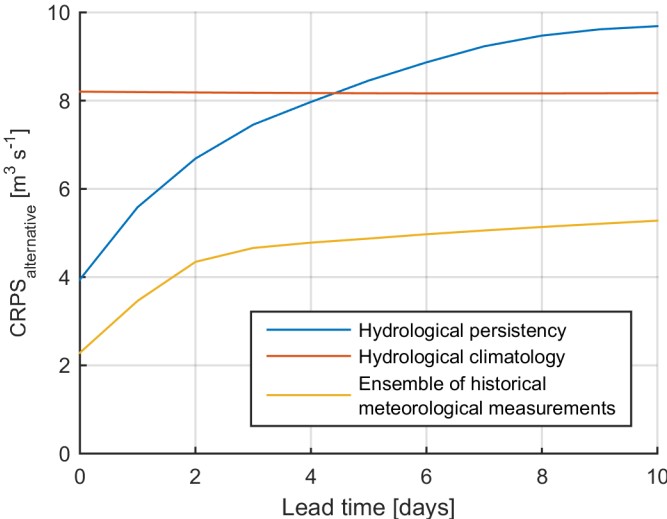

**Figure 3: CRPS of three alternative forecast sets, evaluation period 2008-2013**




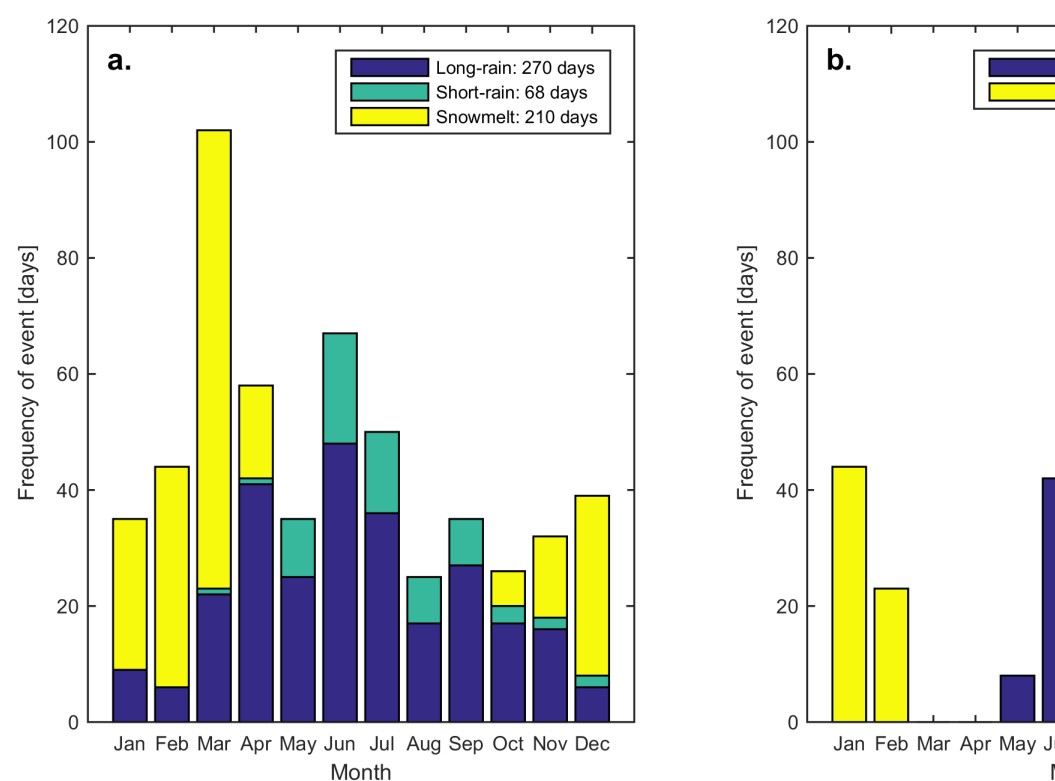

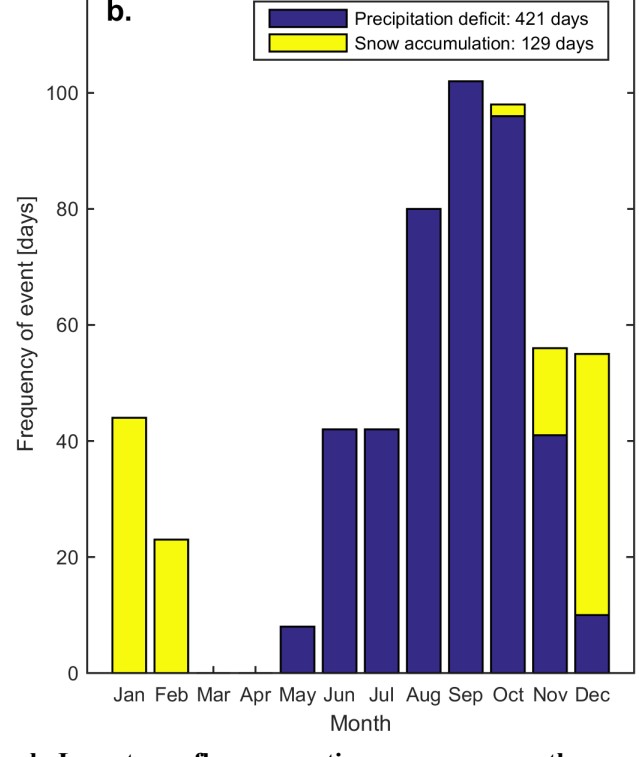

**Figure 4: a. High streamflow generating processes over the year b. Low streamflow generating processes over the year, 1-11-2007 to 31-10-2013**





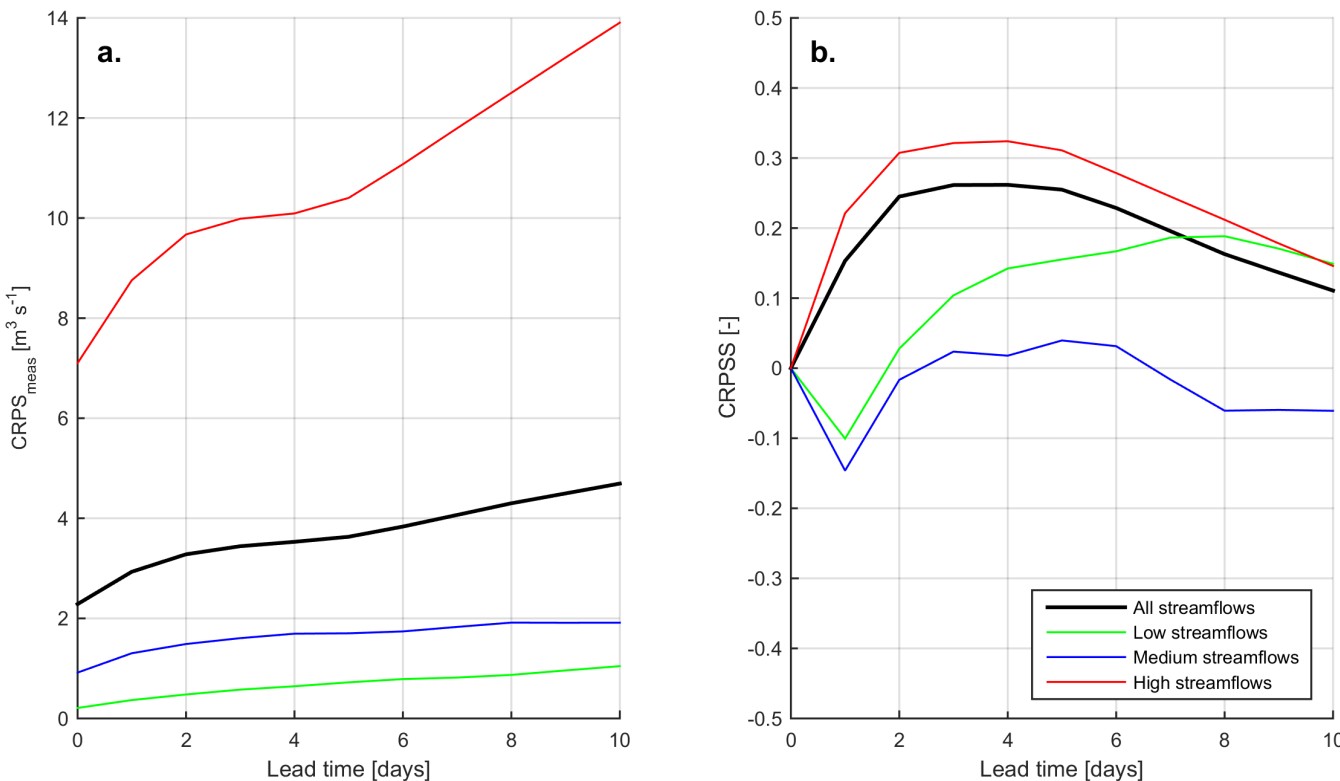

**Figure 5: a. Streamflow forecasts evaluated against streamflow measurements b. Skill of the streamflow forecasts, defined in Eq. 1**

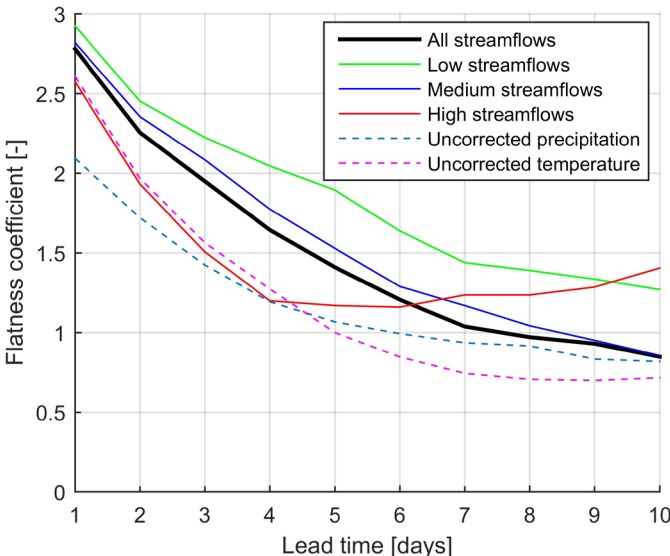

5   **Figure 6: Rank histogram flatness coefficients. The flatness coefficients of the precipitation and temperature forecasts refer to the preceding day.**





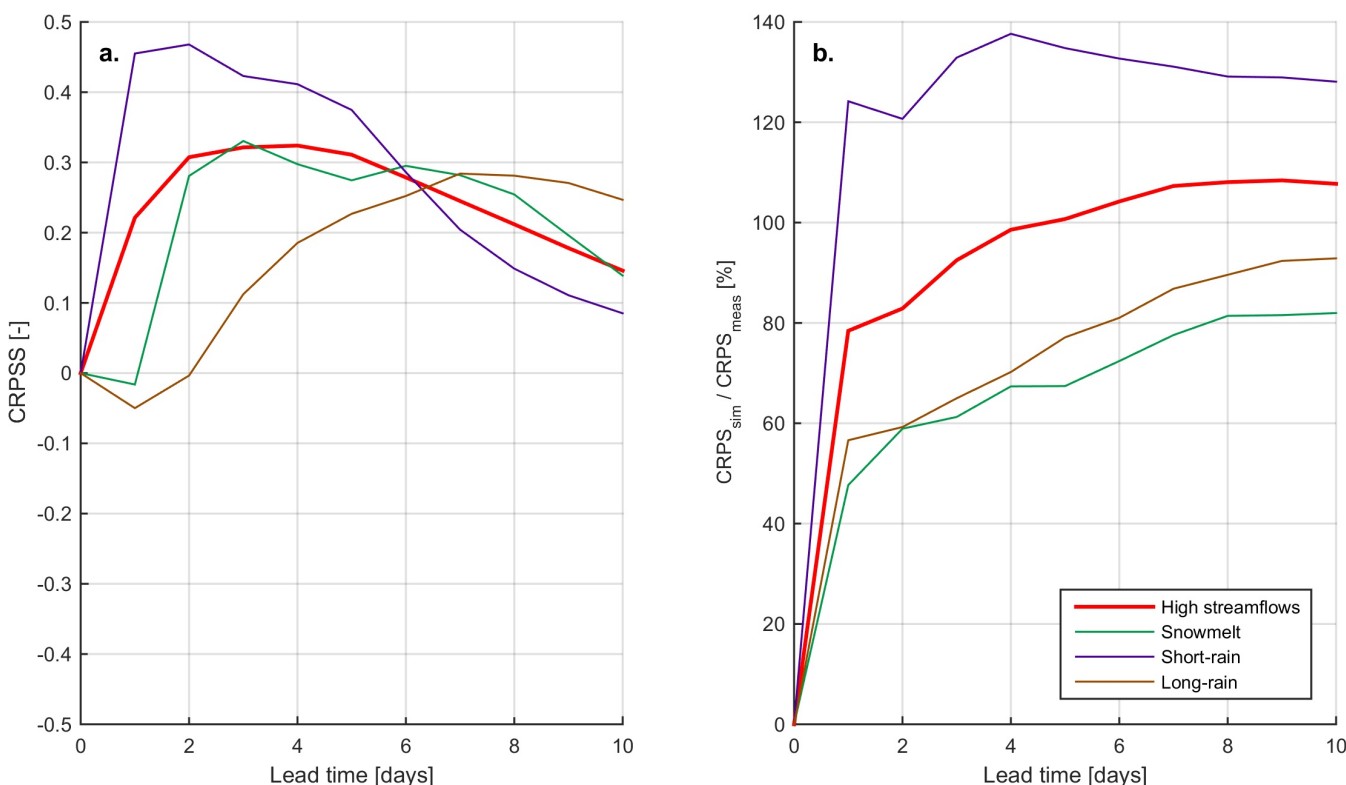

**Figure 7: Ratio of errors in meteorological forecasts (CRPS$_{sim}$) to meteorological forecast + model errors (CRPS$_{meas}$)**

**Figure 8: a. Forecast skill of high streamflow generating processes b. Ratio of errors in meteorological forecasts (CRPS$_{sim}$) to meteorological forecast + model errors (CRPS$_{meas}$).**





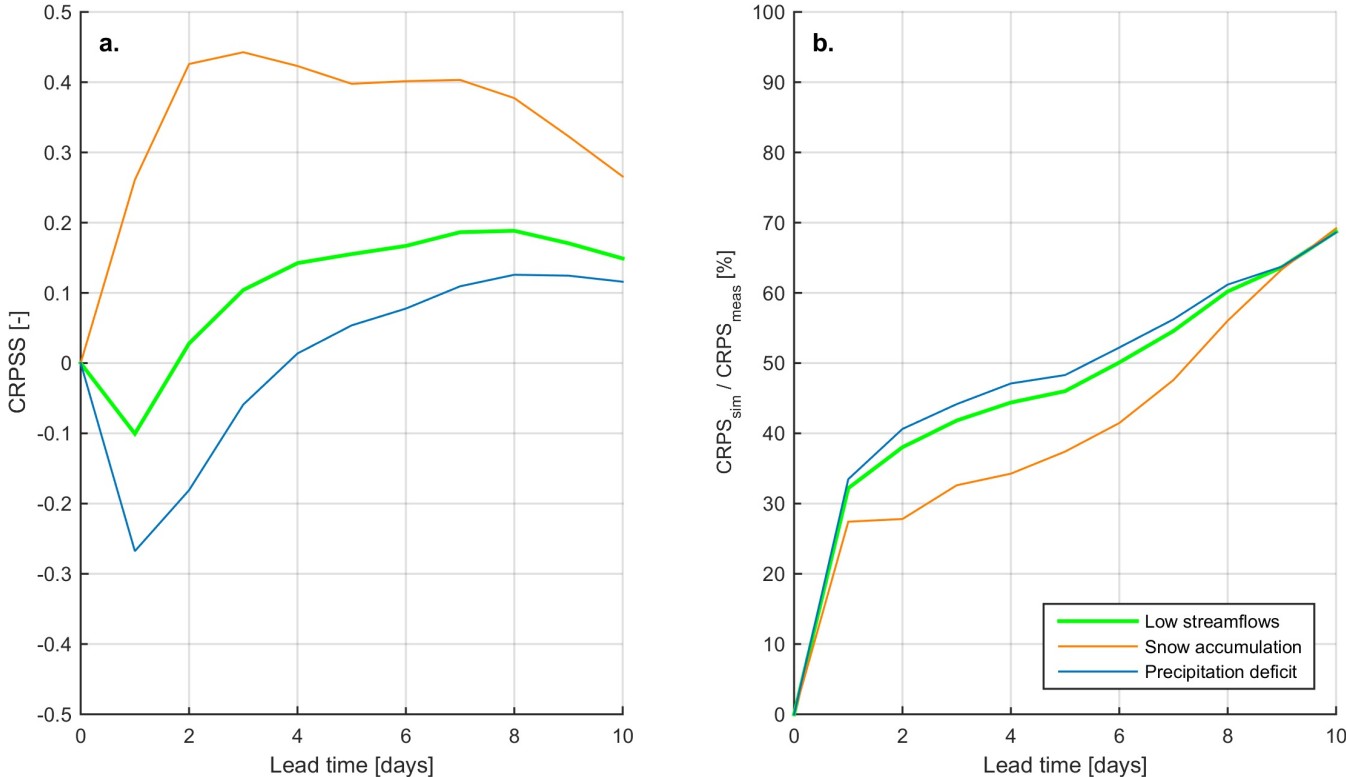

**Figure 9: a. Forecast skill of low streamflow generating processes b. Ratio of errors in meteorological forecasts (CRPS$_{sim}$) to meteorological forecast + model errors (CRPS$_{meas}$).**




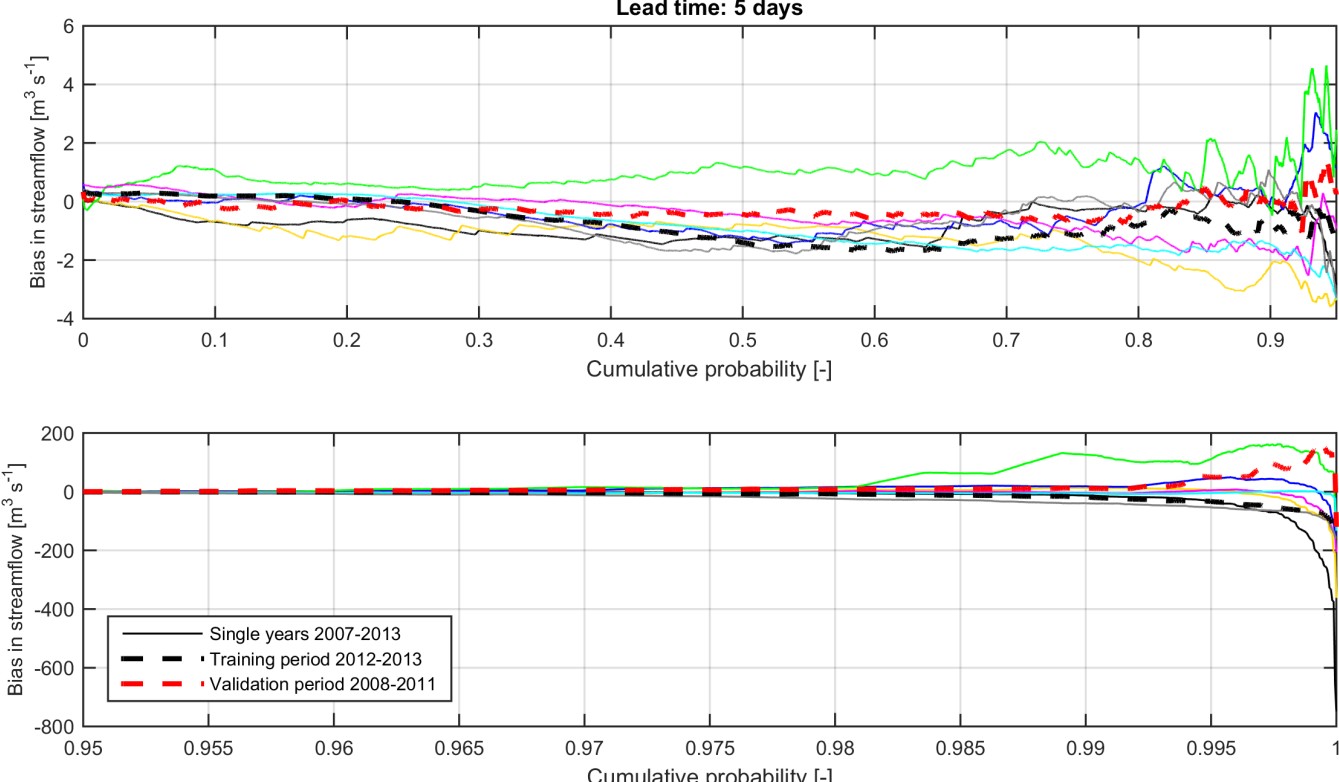

**Figure 10: Difference between CDFs of the measurements and CDFs of the uncorrected streamflow forecasts per hydrological year (upper panel cumulative probability 0 – 0.95 and lower panel 0.95 – 1.0). This figure is for a lead time of 5 days.**

5 **Tables**

**Table 1: Definition of streamflow categories**

| Streamflow category | Thresholds | Streamflow (from measurements 1-11-2007 to 31-10-2013) |
|---|---|---|
| Low streamflow | $Q_{obs} \leq Q_{75}$ | $Q_{obs} \leq 2.76 \ m^3/s$ |
| Medium streamflow | $Q_{75} < Q_{obs} \leq Q_{25}$ | $2.76 \ m^3/s < Q_{obs} \leq 10.35 \ m^3/s$ |
| High streamflow | $Q_{25} < Q_{obs}$ | $10.35 \ m^3/s < Q_{obs}$ |



**Table 2: Characterization of the high streamflow generating processes**

| Process | Characterization | Rules for classification |
|---|---|---|
| Snowmelt flood | Snowmelt floods and rain-on-snow floods (explained by Merz and Blöschl (2003)) are considered as one category. All high streamflow events where snow is involved are characterized as snowmelt floods, because the snowpack and/or frozen soil underneath play an important role in the runoff process. | • Snowpack (HBV) at forecast day-1 |
| Short-rain flood | Short-rain floods and flash floods (characterized by Merz and Blöschl (2003)) are combined. Flash floods are classed in this category as well, because only daily measurements and forecasts are available. | • No snowpack (HBV) at forecast day-1 <br> • Rainfall at forecast day-1 above 10 mm: With small initial storage in the catchment (HBV), precipitation of 10 mm day$^{-1}$ at the day preceding the streamflow event causes a streamflow event above the high streamflow threshold. |
| Long-rain flood | Long-rain flood processes are explained by Merz and Blöschl (2003). This category applies when a streamflow event is not directly generated by snowmelt or high precipitation. | • No snowpack (HBV) at forecast day-1 <br> • Rainfall at forecast day-1 below 10 mm |

**Table 3: Characterization of the low streamflow generating processes**

| Process | Characterization | Rules for classification |
|---|---|---|
| Snow accumulation | If precipitation is snow and does not melt directly, accumulation occurs. | • Snowpack (HBV) at forecast day-1 |
| Precipitation deficit | When low rainfall and high evapotranspiration last over a prolonged period the catchment will dry out. | • No snowpack (HBV) at forecast day-1 |



**Table 4: Calibration and validation performances of the model**

| Calibration run | Calibration (1-11-1971 to 31-10-2000) | | | Validation (1-11-2000 to 31-10-2013) | | |
|---|---|---|---|---|---|---|
| | $Y$ | NS | $E_{RV}$ | $Y$ | NS | $E_{RV}$ |
| Calibration with uncorrected input data | 0.78 | 0.78 | 0% | 0.69 | 0.74 | 6.5% |
| Calibration with input data corrected for elevation (see Sect. 2.1) | 0.81 | 0.81 | 0% | 0.72 | 0.77 | 6.7% |

**Table 5: Performance over the evaluation period 2008-2013, for low, medium and high streamflow simulations (perfect forecasts).**
5 **The initial states are updated at the lead time of 0 days.**

| Lead time [days] | $E_{RV}$ [%] | | | NS [-] | | | $E_{RMA}$ [-] | | |
|---|---|---|---|---|---|---|---|---|---|
| | Low flows | Medium flows | High flows | Low flows | Medium flows | High flows | Low flows | Medium flows | High flows |
| No updating | 43.3 | 7.29 | 1.81 | -10.9 | -2.36 | 0.82 | 0.71 | 0.43 | 0.33 |
| 0 | 3.23 | 4.69 | 2.16 | 0.34 | -0.14 | 0.86 | 0.11 | 0.16 | 0.25 |
| 1 | 6.44 | 7.16 | 2.64 | -0.64 | -0.53 | 0.84 | 0.19 | 0.21 | 0.29 |
| 2 | 8.55 | 8.80 | 2.48 | -1.12 | -0.88 | 0.83 | 0.23 | 0.25 | 0.31 |
| 3 | 11.5 | 9.60 | 2.30 | -2.09 | -1.07 | 0.83 | 0.29 | 0.28 | 0.32 |
| 4 | 13.6 | 10.1 | 2.17 | -2.76 | -1.15 | 0.83 | 0.33 | 0.30 | 0.32 |
| 5 | 15.9 | 10.4 | 2.04 | -3.50 | -1.33 | 0.83 | 0.37 | 0.31 | 0.32 |
| 6 | 18.2 | 10.4 | 1.98 | -4.36 | -1.43 | 0.83 | 0.41 | 0.32 | 0.32 |
| 7 | 19.2 | 10.5 | 2.01 | -4.56 | -1.53 | 0.83 | 0.43 | 0.34 | 0.32 |
| 8 | 20.6 | 10.3 | 2.07 | -4.88 | -1.62 | 0.83 | 0.45 | 0.35 | 0.32 |
| 9 | 22.9 | 10.1 | 2.09 | -5.73 | -1.70 | 0.83 | 0.49 | 0.35 | 0.32 |
| 10 | 24.0 | 10.0 | 2.13 | -6.09 | -1.77 | 0.83 | 0.50 | 0.36 | 0.32 |