# Peer review of "Performance of ensemble streamflow forecasts under varied hydrometeorological conditions"

_Hydrology and Earth System Sciences, 2016_

## Short Comment (SC1) · 28 Nov 2016

At page 2, lines 3-5, where authors assess "Uncertainties in streamflow forecasts originate from meteorological input, and hydrological model parameters, initial conditions and model structure", they could also cite:

Biondi, D., De Luca, D.L. (2012). A Bayesian approach for real-time flood forecasting. Physics and Chemistry of the Earth, 42-44, pp. 91-97. DOI: 10.1016/j.pce.2011.04.004

Biondi, D., De Luca, D.L. (2013).Performance assessment of a Bayesian Forecasting System (BFS) for real-time flood forecasting.Journal of Hydrology, 479, pp. 51-63. DOI: 10.1016/j.jhydrol.2012.11.019

---

## Referee Comment (RC1) · Anonymous Referee #1 · 16 Dec 2016

General comments

This manuscript presents an interesting analyse of the performance of hydrological ensemble predictions. The skills are screened according the regime (low and high streamflow) and the generating processes (snow melt, short rain, long rain floods etc.). This study further disentangles hydrological model errors and errors from meteorological forcing. The methodology is applied to a mountainous catchment. The combination of existing methodologies is pertinent and is worth being published in HESS.

However the reading is not easy and a major revision is necessary. Some information is redundant in the introduction, methodology and results sections and long lists of references are not always necessary. The focus should be made on the main contribution

of the paper i.e. the analysis of the skill for different hydro-meteorological conditions and skip or shorten secondary experiments. Some validation methodologies are described but their results are not shown. A balance should be found: either shorten the description or include those results. Some suggestions are given in the specific comments. The English should be improved.

The authors are using ensemble predictions from ECMWF from 2007 to 2013 with a training of the pre- and post-processing during two water years between 2011 and 2013. They associate the failure of the quantile mapping for post-processing method to the short time series of forecasts for training and to the inconsistency of the bias between the training and the validation period. They forget that the ensemble prediction system has undergone many changes during this period including spatial resolution changes. This is why retrospective forecasts are available since long and provide samples of 18 to 20 years back for post-processing purposes. Re-forecasts have been widely used and reported in the literature. These meteorological re-forecasts have also been used for the preparation of hydrological re-forecasts for the statistical post-processing of hydrological ensemble predictions.

Figure 5 to 9 are the core of the paper. They will gain value if the plots are associated with confidence intervals.

The use of the term "perfect forecast" is questioned because it is neither a forecast nor perfect and, would the future meteorological forcing be known, predictions with the model would include growing errors due to initial conditions as somehow shown in Table 5.

Specific comments

P1, L20-24 Should be rephrased e.g. too many occurrence of "improve".

P3 L23-P4, L3 How do you correct measurement? Do you correct each station for the difference between the elevation of the station and the average of the elevation

in the area defined by the intersection of the Thiessen polygon corresponding to the station and the watershed? Then average the corrected values of the stations using their relative contribution to the catchment area as weights?

P5, L20-21 Equations would be appropriate here in order to define Y, NS and E_RV.

P5, L28 preceding the first forecast day.

P5, L32-P6, L2 I would suggest to skip this experiment or, if impossible to skip, tell already that it failed (according to P11, L3-5). This is to lighten the methodologies to keep in mind until the result section.

P6, L31-P7, L11 Some information (and references) is redundant with the sub-sections.

P7, L1 Three properties of probabilistic forecast quality . . .

P7, L8 "The histograms accompanying ..." the histograms of what?

P7, L20-21 "CRPS approaches the average value of the evaluated variable" What do you mean with "approaches"?

P7, L24-27 "and compares the forecasts with a relevant alternative forecast" somehow redundant with the beginning of the sentence.

P8, L1-2 "... argue that this" choice . . . these two lines should be rephrased. I would prefer a positive phrasing saying that the choice of another alternative forecast may result in a more robust estimation of forecast skill.

P8, L22-23 Either provide an equation for the "numerical indicator delta" if it adds to the understanding of the adopted methodology or skip any reference to delta.

P8, L30-31 "... contain a random element ..." explain how it works for the flatness coefficient.

P9, L3 "... for a certain event ..." It would be useful to define "event" and refer to sub-section 3.4 or Table 1.
P9, L24-28 Almost the same thing is repeated.

P9, L29 At a first reading, it was tempting to replace this ratio with a CRPSS of sim against meas but the purpose is different and since it is a major tool in this paper, this paragraph should be written with much care.

P10, L11-12 Are the rules given also by Merz and Blöschl or defined for this catchment based for instance on data from both simulation and observations during the training period?

P10, L16 Do you mean that the distribution of the generating processes shown in the figure is like we can expect for this region?

P10, L19, Table 3 What is the rule for precipitation deficit?

P11, L21 "preceding day" the day before the forecast issuing day.

P11, L28-29 "not shown in the paper" therefore, going back to section 3.1.3, the methodology description should be simpler and not encumber with strategy numbers.

P10 L20 What do you mean by "reliable distribution"?

P12, L13 with more skill instead of "skilful"

P12, L16 "functional" what do you mean?

P12, L28 "... are in general less predictable by historical measurements ..." please re-phrase

P12, L32 "not shown" a figure is missing with the rank histograms for the low streamflow forecasts and for the high streamflow forecasts, two lead times. Apparently, for high flow, the rank histogram is not exactly U-shaped but skewed according to P13, L12-13.

P13 L10-13 Difficult to figure out ... Please add a figure with the reliability diagrams and corresponding sharpness histograms for the low streamflow forecasts and for the high streamflow forecasts two lead times.

P13 L15-17 Note that good sharpness without reliability is useless.

P13, L18 reference already given, please re-phrase.

P14, L11-13 "... the below zero skill . . . do not result in positive skill ..." ...

P14, L23 What is the amount of this fake drizzle?

P14, L24-26 Re-phrase: "... meteorological forecasts accumulated in the forecasting system are better model inputs ..."

P15, L8 & Figure 10 I would skip this figure which highlights the weakness of drawing such a detailed profile with just a water-year data. The legend is missing for the thin plain lines.

P16, L8-10 Do you have evidence that such coincidence occurs and is the main explanation for the high ratio for short-rain floods?

P17, L13-15 "longer time series of forecasts", "longer forecasts datasets" see general comments; "more sophisticated" and first of all more robust.

---

## Referee Comment (RC2) · Anonymous Referee #2 · 15 Feb 2017

General comments:

This paper summarizes the application of the widely used HBV hydrologic model to streamflow forecasting in a Polish mountain river. The project uses ECMRWF ensemble weather forecasts to drive the streamflow model, and explores both pre- and post-processing of the ensembles for bias correction. Useful results are obtained, and the study has significant potential. I recommend that the paper is accepted pending major revisions.

Detailed comments:

1. The paper repeatedly refers to HBV as a spatially lumped model. This isn't just

terminology, as around lines 20-25 of page 15, the manuscript seems to imply that the model assumes a snowpack to be present (or absent) across the entire model domain. There are a few versions of HBV, but it's normally viewed as semi-distributed, using (at a minimum) elevation bands.

2. The manuscript makes a good point on lines 29-30 of page 1 about socio-economic development increasing the impacts of extreme hydrometeorological events. It also probably bears mentioning that climate changes, both natural and anthropogenic, may further exacerbate these impacts. See Perkins, Pagano, and Garen, "Innovative operational seasonal water supply forecasting technologies," Journal of Soil and Water Conservation, 2009; and Fleming, "Demand modulation of water scarcity sensitivities to secular climatic variation: theoretical insights from a computational maquette," Hydrological Sciences Journal, 2016.

3. Terms could stand to a little better defined. For example, most flood and water supply forecasters who I know would regard "short-term" forecasts as having lead times of 0-10 days, and "long-term" forecasts as having lead times of weeks to months. So what the authors refer to here as "medium-term" would be referred to as "short-term" by many if not most others working in the field. And no effort is made here to distinguish medium-term from short-term hydrologic forecasting. More broadly, some of the wording throughout the manuscript would benefit from a re-think for better clarity and precision.

4. Why is only meteorological forecast uncertainty incorporated into the ensemble model? It's commonplace in the research literature for forecast models to include both meteorological uncertainty (NWP ensemble) and hydrologic model parameterization uncertainty (ensemble of hydrologic parameter values). This work is starting to make its way into operational practice too. Providing some justification for this choice might be a good idea.

5. The description of the model implementation isn't quite adequate. What was the

calibration-testing split, and what were the model performances during both phases? And it's stated that the objective function selected for calibration is "Y", which apparently combines the Nash-Sutcliffe efficiency with a volumetric error measure. Objective function selection is a key step in model calibration, and more information needs to be provided, starting with an explicit mathematical definition for "Y".

6. The updating of initial states was performed here for the slow-runoff and fast-runoff reservoirs. That's interesting and useful, but why was SWE not selected as the object of this data assimilation exercise? It seems like it would be a more rewarding, and certainly more conventional, choice in this northern continental European mountain catchment.

7, The literature review of ensemble hydrologic forecasting, pre- and post-processing for bias corrections, and data assimilation and model updating, is a good start but seems a little light. Citing more work would provide valuable context to the paper. A reasonable place to start might be recent work by Dominique Bourdin at the University of British Columbia and Hamid Moradkhani at Portland State University.

8. Some of the specific conclusions seem a little surprising. That's great, but it also means they'd benefit from additional discussion. In particular, the paper concludes in section 4.2.1 that the quality of the forecasts at lead times of less than 3 days is dominated by hydrologic initial conditions, and the weather forecasts become the dominant source of predictive skill after that. This would be a reasonable conclusion for a large or flat basin, but for a small, steep mountain river it seems a little surprising – these are typically flashy systems that respond to rain or snowmelt inputs within a day or so. Indeed, a few pages later near the end of section 5, the paper states that "in the hydrological model the lag time between a rainfall event and the streamflow peak is set to 1 day." It also seems that conclusions like this, which attempt to attribute predictive skill (and therefore also predictive error) to various different sources, might be difficult to make convincingly without using a more statically sophisticated and exhaustive data assimilation procedure, incorporating ensembles of hydrologic models and/or model

parameters, etc.

---

## Referee Comment (RC3) · Anonymous Referee #3 · 15 Feb 2017

The authors proposed a methodology to give insight in the performance of ensemble streamflow forecasting systems in three streamflow categories (low, medium and high) and related runoff generating processes from lead times of 1 day to 10 day with a case study in a mountainous river catchment of less than 1000 sqr km in Poland. The quantitative precipitation forecasts and temperature forecasts extracted from the European Centre for Medium-Range Weather Forecasts (ECMWF) are averaged with catchment as input of a lumped hydrological (HBV) to generate ensemble streamflow. Several intensively used verification measures (CRPS, CRPSS, Rank histogram, Reliability diagram and ROC) are selected to evaluate the ensemble forecasts. Additionally, the pre-processing, post-processing and updating of model initial states are adopted to

improve the behavior of the system.

Generally speaking, the study gave an interesting investigation on the assessment of hydrological ensemble prediction system on different runoff processes including snowmelt, short-rain flood and so on, and a further analysis was made on the uncertainty source of these varied hydrometeorological conditions. There I suggest accept this manuscript after a moderate revision.

There are a few issues list below that the authors should address: 1) The logic in Paragraph 2 and 3 of Section 1 needs to be perfect. Some irrelevant statements can be removed, eg. SOME CONTENTS from Line 10 to Line 15 in Page 2 about EFAS are unnecessary to some degree. 2) Lines18-20 Page 6: A further explanation is expected why the training period is defined from 2011-2013 while the years previous to 2011 is used to validation. 3) In Section 3.2, it is not necessary to introduce all the evaluation scores in details, for the CRPS, CRPSS, Reliability diagram and ROC can be regarded as "industry standards" in ensemble forecasting, so simply citing the relevant references. 4) In Section 4.1.2, it is confusing that since the QM pre-processing brings improvement to the precipitation and temperature forecasts, why the conclusion is that the strategy 0 results in the best CRPS. 5) The figures about rank histograms and reliability diagrams are missing or not shown intentionally? 6) The catchment area is less than 1000km2 and the data used are daily. For flood forecasting in such catchment area, is it daily data too coarse? Perhaps 3h or 6h subdaily data are more useful for flood forecasting in such area. Please make it an elaborate story. 7) For flood forecasting, flood peak, volume and peak time are all important. Can these be analyzed in the study? 8) Page 9: It is not very clear how the errors are contributed in Section 3.3. Why can CRPSsim/CRPSmeans represent the error contribution? Please add more details.

---

## Author Comment (AC1) · 18 Mar 2017

**Response to Interactive comment Anonymous Referee #1**

*General comments*

**Comment:** *This manuscript presents an interesting analyse of the performance of hydrological ensemble predictions. The skills are screened according the regime (low and high streamflow) and the generating processes (snow melt, short rain, long rain floods etc.). This study further disentangles hydrological model errors and errors from meteorological forcing. The methodology is applied to a mountainous catchment. The combination of existing methodologies is pertinent and is worth being published in HESS.*

*However the reading is not easy and a major revision is necessary. Some information is redundant in the introduction, methodology and results sections and long lists of references are not always necessary. The focus should be made on the main contribution of the paper i.e. the analysis of the skill for different hydro-meteorological conditions and skip or shorten secondary experiments. Some validation methodologies are described but their results are not shown. A balance should be found: either shorten the description or include those results. Some suggestions are given in the specific comments. The English should be improved.*

**Reply:** We thank the reviewer for the assessment. We appreciate the reviewer's opinion about the study and the valuable suggestions provided to improve the manuscript. Below are our responses to the comments and points raised.

The reviewer's suggestion to improve the flow of the paper is valuable, and the specific comments contain many relevant points for this.

With respect to the comment to increase the focus of the paper on the main scientific innovation, we will leave out the additional updating experiment, which has also not been used because it was unsuccessful (P5 Line 31 – P6 Line 2, P11 Line 3 – P11 Line 5).

Regarding the experiments on pre- and post-processing of the ensemble forecasts we consider this important and propose not to remove it from the paper. The procedure is common, so removing it will presumably result in doubts about why we have not applied a correction procedure. The results of this experiment are quite striking and we will add a figure with CRPS values for the different pre- and post-processing strategies showing this finding (see Figure 1).

Further replies to this comment follow below in response to the specific comments.

[Figure]

*Figure 1: CRPS of the post-processing strategies over the validation period 2008-2011*

**Comment:** *The authors are using ensemble predictions from ECMWF from 2007 to 2013 with a training of the pre- and post-processing during two water years between 2011 and 2013. They associate the failure of the quantile mapping for post-processing method to the short time series of forecasts for training and to the inconsistency of the bias between the training and the validation period. They forget that the ensemble prediction system has undergone many changes during this period including spatial resolution changes. This is why retrospective forecasts are available since long and provide samples of 18 to 20 years back for post-processing purposes. Re-forecasts have been widely used and reported in the literature. These meteorological re-forecasts have also been used for the preparation of hydrological re-forecasts for the statistical postprocessing of hydrological ensemble predictions.*

**Reply:** It is correct that we used meteorological forecasts from a system that has undergone changes. The TIGGE data portal contains the operational forecasts from meteorological forecast centres. We agree that this affects the pre-processing and post-processing results and we thank the reviewer for this suggestion. We will add a statement to Page 15 Line 15-16 that the joint distribution of measurements and forecasts is nonhomogeneous in time, because the meteorological forecast system has undergone changes during our analysis period (Mladek, 2016):

https://software.ecmwf.int/wiki/display/TIGGE/Model+upgrades#Modelupgrades-ECMWF

**Comment:** *Figure 5 to 9 are the core of the paper. They will gain value if the plots are associated with confidence intervals.*

**Reply:** CRPS and CRPSS are the main evaluation scores that we used. In recent literature these scores are commonly applied without associated confidence intervals or statistical tests, by Demargne et al. (2010), Hersbach (2000), Pappenberger et al. (2015), Renner et al. (2009), Verkade et al. (2013), and Ye et al. (2014). We agree to the suggestion that confidence intervals around the CRPS values would add value to the figures, but we consider establishing such confidence intervals outside the focus of this paper.

*Comment: The use of the term "perfect forecast" is questioned because it is neither a forecast nor perfect and, would the future meteorological forcing be known, predictions with the model would include growing errors due to initial conditions as somehow shown in Table 5.*

**Reply:** We appreciate the comment. The term "perfect forecast" was introduced by Olsson and Lindström (2008), but the term is somewhat misleading. For the same concept, Renner et al. (2009) used the term "baseline simulation", Demargne et al. (2010) used the term "simulated flow", Verkade et al. (2013) used the term "simulated streamflow" and Bennett et al. (2014) used the term "perfect-rainfall-forced forecasts". We propose to use "observed meteorological input forecasts".

*Specific comments*

**Comment:** *P1, L20-24 Should be rephrased e.g. too many occurrence of "improve".*

**Reply:** We agree to the comment and will change it to:

"To improve the performance of the forecasting system for high streamflow events,  the meteorological forecasts are crucial. For low streamflow forecasts, it is advised to calibrate a hydrological model specifically on low streamflow events. The study further recommends improving the reliability of the ensemble streamflow forecasts, by including the uncertainties in hydrological model parameters and initial conditions, and by increasing the dispersion of the meteorological input forecasts."

**Comment:** *P3 L23-P4, L3 How do you correct measurement? Do you correct each station for the difference between the elevation of the station and the average of the elevation in the area defined by the intersection of the Thiessen polygon corresponding to the station and the watershed? Then average the corrected values of the stations using their relative contribution to the catchment area as weights?*

**Reply:** The assumption of the reviewer is correct: this is the procedure that we used. We will revise the text to make this clear:

"Precipitation, temperature and streamflow measurement series are available at a daily time interval for the period 1 January 1971 to 31 October 2013, provided by the Polish Institute of Meteorology and Water Management. Precipitation and temperature data from 5 measurement stations (Fig. 1) have been selected because of their distribution over the catchment and data series completeness.  Given that stations are mostly located in valleys and precipitation and temperature vary with elevation, the catchment averages  may be biased (Panagoulia, 1995; Sevruk, 1997). Following Akhtar et al. (2009), precipitation measurements are corrected using relative correction factors (in %), whereas temperature measurements are corrected using absolute correction factors (in °C). The precipitation correction factor differs considerably between months. For December–February the mean precipitation gradient is 10.5 % 100 m$^{-1}$, while for March–November the mean precipitation gradient is 5.4 % 100 m$^{-1}$. Although the number of stations is limited to accurately determine precipitation and temperature gradients, the calculated precipitation gradients are used because of the clear difference between two periods. The temperature gradient does not vary much over the year and therefore the global standard temperature lapse rate of 0.65 °C 100 m$^{-1}$ is applied. The corrected  measurements are weighted based on the relative coverage of Thiessen polygons (Fig. 1), to represent catchment averages. By the corrections the annual mean precipitation increases from 741.2 mm to 768.4 mm and the annual mean potential evapotranspiration decreases from 695.3 mm to 674.4 mm."

*Comment: P5, L20-21 Equations would be appropriate here in order to define Y, NS and E_RV.*

**Reply:** We hesitate to add the equations since Y, NS and $E_{RV}$ are defined in the given references.

*Comment: P5, L28 preceding the first forecast day.*

**Reply:** We agree. We will change it to: "the day preceding the forecast issuing day" (from comment on P11, L21).

*Comment: P5, L32-P6, L2 I would suggest to skip this experiment or, if impossible to skip, tell already that it failed (according to P11, L3-5). This is to lighten the methodologies to keep in mind until the result section.*

**Reply:** We agree to leave this out. Also see the response to the first general comment.

*Comment: P6, L31-P7, L11 Some information (and references) is redundant with the sub-sections.*

**Reply:** We agree. Also looking at comment 3 by Reviewer 3 we will omit general information about the evaluation scores, but focusing on what aspect on forecast quality each score evaluates and citing the relevant references.

*Comment: P7, L1 Three properties of probabilistic forecast quality …*

**Reply:** We do not understand this comment.

*Comment: P7, L8 "The histograms accompanying ..." the histograms of what?*

**Reply:** We will change this sentence to:

" To evaluate sharpness we use histograms which show the sample size over the range of forecast probability bins used to establish the reliability diagrams."

*Comment: P7, L20-21 "CRPS approaches the average value of the evaluated variable" What do you mean with "approaches"?*

**Reply:** We will change "approaches" to "converges to".

*Comment: P7, L24-27 "and compares the forecasts with a relevant alternative forecast" somehow redundant with the beginning of the sentence.*

**Reply:** We will change this sentence to:

"

To eliminate the magnitude of the investigated variable we normalize the CRPS against the CRPS of a relevant alternative forecast, a principle which is also used by Bennett et al. (2014), Demargne et al. (2010), Renner et al. (2009), Velázquez et al. (2010) and Verkade et al. (2013) to evaluate forecast skill."

*Comment: P8, L1-2 "... argue that this" choice … these two lines should be rephrased. I would prefer a positive phrasing saying that the choice of another alternative forecast may result in a more robust estimation of forecast skill.*

**Reply:** We propose to delete P7 Line 30 – P8 Line 2, because it is not really relevant to explain the procedure that we followed. It explains why we have not applied hydrological persistency or

hydrological climatology as alternative forecast set, but we can focus the text on what we have done: using the forecast set with the lowest CRPS values as alternative forecast set, because this set is most difficult to beat in performance.

**Comment:** *P8, L22-23 Either provide an equation for the "numerical indicator delta" if it adds to the understanding of the adopted methodology or skip any reference to delta.*

**Reply:** We will remove the reference to delta.

**Comment:** *P8, L30-31 "... contain a random element ..." explain how it works for the flatness coefficient.*

**Reply:** We will add further explanation about the random element:

"In this case a random rank is assigned from the set of ensembles and the measurement that have the same value."

**Comment:** *P9, L3 "... for a certain event ..." It would be useful to define "event" and refer to sub-section 3.4 or Table 1.*

**Reply:** We will specify "certain event" as "for low streamflow events and high streamflow events (defined in Sect. 3.4)".

**Comment:** *P9, L24-28 Almost the same thing is repeated.*

**Reply:** We will delete P9 Line 24-26.

**Comment:** *P9, L29 At a first reading, it was tempting to replace this ratio with a CRPSS of sim against meas but the purpose is different and since it is a major tool in this paper, this paragraph should be written with much care.*

**Reply:** We will add the equation below (see also comment 8 by Reviewer 3):

$$\frac{CRPS_{sim}}{CRPS_{meas}} \sim \frac{meteorological\ forecast\ errors}{meteorological\ forecast\ errors+hydrological\ model\ errors}$$

If this ratio is low, the hydrological model errors are dominant and if this ratio is high, the meteorological forecast errors are dominant.

**Comment:** *P10, L11-12 Are the rules given also by Merz and Blöschl or defined for this catchment based for instance on data from both simulation and observations during the training period?*

**Reply:** The study by Merz and Blöschl (2003) is used to characterize the high streamflow generating processes in Table 2. The rules for classification are defined specifically for the study catchment and are based on observations and model simulations. We will change the text to:

"Various runoff contributing processes can result in high flows. Table 2 defines the processes and  rules for classification . The rules for classification are based on rainfall observations and snowpack model simulations; at one day before the event because of the time step of the HBV model."

**Comment:** *P10, L16 Do you mean that the distribution of the generating processes shown in the figure is like we can expect for this region?*

**Reply:** The reviewer's interpretation is correct. We will change this to:

"Figure 4a presents the distribution of high streamflow generating processes over the year following the  rules for classification in Table 2.

 The distribution of processes over the year is like we can expect for this region."

Likewise we will change P10 Line 20-21.

*Comment: P10, L19, Table 3 What is the rule for precipitation deficit?*

**Reply:** The rule used for classifying an event as a precipitation deficit generated low streamflow is that if there is a low streamflow event and if there is no snowpack present (based on model simulations) we assume that the low streamflow event is caused by a precipitation deficit. We think that the definition in Table 3 is clear.

*Comment: P11, L21 "preceding day" the day before the forecast issuing day.*

**Reply:** We agree and we will change "preceding day" to "day preceding the forecast issuing day" accordingly in the paper (also see comment P5 Line 28).

*Comment: P11, L28-29 "not shown in the paper " therefore, going back to section 3.1.3, the methodology description should be simpler and not encumber with strategy numbers.*

**Reply:** This comment is discussed in the response to the first comment.

*Comment: P10 L20 What do you mean by "reliable distribution"?*

**Reply:** See response to comment P10, L16.

*Comment: P12, L13 with more skill instead of "skilful"*

**Reply:** Skill is defined as the performance of the streamflow forecast relative to the performance of alternative forecasts. Here we do not mean 'with more skill', but skilful relative to the alternative forecasts.

*Comment: P12, L16 "functional" what do you mean?*

**Reply:** We will change "functional" to "plausible".

*Comment: P12, L28 "... are in general less predictable by historical measurements ..." please re-phrase*

**Reply:** P12 Line 28-29 is partly a repetition of the preceding sentence, so this sentence will be deleted. We will change P12 Line 27-29 to:

"In addition,  high streamflow events will be less well captured  by historical measurements, and thus  the alternative forecasts will have lower quality for these events. "

*Comment: P12, L32 "not shown" a figure is missing with the rank histograms for the low streamflow forecasts and for the high streamflow forecasts, two lead times. Apparently, for high flow, the rank histogram is not exactly U-shaped but skewed according to P13, L12-13.*

**Reply:** To keep the paper short we chose not to include these figures in the paper. However, we think that the results are relevant and therefore we described them in words. We agree that this makes reading of the paper difficult and the results nontransparent. We could make the figures available by a supplement to the paper.

*Comment: P13 L10-13 Difficult to figure out ... Please add a figure with the reliability diagrams and corresponding sharpness histograms for the low streamflow forecasts and for the high streamflow forecasts two lead times.*

**Reply:** See response to comment P12 L32.

*Comment: P13 L15-17 Note that good sharpness without reliability is useless.*

**Reply:** We agree. We will emphasize this in the conclusion (bullet 1).

*Comment: P13, L18 reference already given, please re-phrase.*

**Reply:** We agree. We will change this to:

"All AUC values are above 0.85,  which indicates a good resolution of the streamflow forecast system."

*Comment: P14, L11-13 "… the below zero skill … do not result in positive skill …"*

**Reply:** We agree that this sentence is not well written. We will change the sentence to:

"

For long-rain floods and snowmelt floods, the meteorological forecasts at small lead times do not result in positive skill as compared to forecasts based on historical meteorological measurements."

*Comment: P14, L23 What is the amount of this fake drizzle?*

**Reply:** This is an interesting question, but we consider this to be out of the focus of this paper.

*Comment: P14, L24-26 Re-phrase: "... meteorological forecasts accumulated in the forecasting system are better model inputs ..."*

**Reply:** We agree that this sentence is not well written. We will change the sentence to:

"The skill increases for larger lead times, so for larger lead times ECMWF meteorological forecasts accumulated in the forecasting system give better predictions than historical measurements ."

*Comment: P15, L8 & Figure 10 I would skip this figure which highlights the weakness of drawing such a detailed profile with just a water-year data. The legend is missing for the thin plain lines.*

**Reply:** We hesitate to skip this figure, because it illustrates why the pre- and post-processing procedures are not working: the training period and validation period show different bias distributions, because of the short time series.

The thin plain lines are showed in the legend as "Single years 2007-2013". We will add an explanation to the caption that each thin line refers to a single year between 2007 and 2013.

*Comment: P16, L8-10 Do you have evidence that such coincidence occurs and is the main explanation for the high ratio for short-rain floods?*

**Reply:** This is an interesting question and we will investigate how often this occurs.

*Comment: P17, L13-15 "longer time series of forecasts", "longer forecasts datasets" see general comments; "more sophisticated" and first of all more robust.*

**Reply:** We had to deal with the limitations of available data and to focus on the objective of the study we made choices in the development of the ensemble forecasting system. In the responses to the

general comments and in the responses to Reviewer 2 and Reviewer 3 these choices are further explained.

**References**

Akhtar, M., Ahmad, N. and Booij, M. J.: Use of regional climate model simulations as input for hydrological models for the Hindukush-Karakorum-Himalaya region, Hydrol. Earth Syst. Sci., 13(7), 1075–1089, doi:10.5194/hess-13-1075-2009, 2009.

Bennett, J. C., Robertson, D. E., Shrestha, D. L., Wang, Q. J., Enever, D., Hapuarachchi, P. and Tuteja, N. K.: A System for Continuous Hydrological Ensemble Forecasting (SCHEF) to lead times of 9 days, J. Hydrol., 519, 2832–2846, doi:10.1016/j.jhydrol.2014.08.010, 2014.

Demargne, J., Brown, J., Liu, Y., Seo, D. J., Wu, L., Toth, Z. and Zhu, Y.: Diagnostic verification of hydrometeorological and hydrologic ensembles, Atmos. Sci. Lett., 11(2), 114–122, doi:10.1002/asl.261, 2010.

Hersbach, H.: Decomposition of the Continuous Ranked Probability Score for Ensemble Prediction Systems, Weather Forecast., 15(5), 559–570, doi:10.1175/1520-0434(2000)015<0559:DOTCRP>2.0.CO;2, 2000.

Merz, R. and Blöschl, G.: Regional flood risk - what are the driving processes?, in Water Resources Systems-Hydrological Risk, Management and Development, edited by G. Blöschl, S. Franks, M. Kumagai, K. Musiake, and D. Rosbjerg, pp. 49–58, International Association of Hydrological Sciences Press, Wallingford, UK. [online] Available from: http://hydrologie.org/redbooks/a281/iahs_281_049.pdf, 2003.

Mladek, R.: Model upgrades, TIGGE [online] Available from: https://software.ecmwf.int/wiki/display/TIGGE/Model+upgrades#Modelupgrades-ECMWF (Accessed 7 March 2017), 2016.

Olsson, J. and Lindström, G.: Evaluation and calibration of operational hydrological ensemble forecasts in Sweden, J. Hydrol., 350(1–2), 14–24, doi:10.1016/j.jhydrol.2007.11.010, 2008.

Panagoulia, D.: Assessment of daily catchment precipitation in mountainous regions for climate change interpretation, Hydrol. Sci. J., 40(3), 331–350, doi:10.1080/02626669509491419, 1995.

Pappenberger, F., Ramos, M. H., Cloke, H. L., Wetterhall, F., Alfieri, L., Bogner, K., Mueller, A. and Salamon, P.: How do I know if my forecasts are better? Using benchmarks in hydrological ensemble predictions, J. Hydrol., 522, 697–713, doi:10.1016/j.jhydrol.2015.01.024, 2015.

Renner, M., Werner, M. G. F., Rademacher, S. and Sprokkereef, E.: Verification of ensemble flow forecasts for the River Rhine, J. Hydrol., 376(3–4), 463–475, doi:10.1016/j.jhydrol.2009.07.059, 2009.

Sevruk, B.: Regional dependency of precipitation-altitude relationship in the Swiss Alps, Clim. Change, 36(3–4), 355–369, doi:10.1023/A:1005302626066, 1997.

Velázquez, J. A., Anctil, F. and Perrin, C.: Performance and reliability of multimodel hydrological ensemble simulations based on seventeen lumped models and a thousand catchments, Hydrol. Earth Syst. Sci., 14(11), 2303–2317, doi:10.5194/hess-14-2303-2010, 2010.

Verkade, J. S., Brown, J. D., Reggiani, P. and Weerts, A. H.: Post-processing ECMWF precipitation and temperature ensemble reforecasts for operational hydrologic forecasting at various spatial scales, J. Hydrol., 501, 73–91, doi:10.1016/j.jhydrol.2013.07.039, 2013.

Ye, J., He, Y., Pappenberger, F., Cloke, H. L., Manful, D. Y. and Li, Z.: Evaluation of ECMWF medium-range ensemble forecasts of precipitation for river basins, Q. J. R. Meteorol. Soc., 140(682), 1615–1628, doi:10.1002/qj.2243, 2014.

---

## Author Comment (AC2) · 18 Mar 2017

**Response to Interactive comment Anonymous Referee #2**

*General comments:*

*This paper summarizes the application of the widely used HBV hydrologic model to streamflow forecasting in a Polish mountain river. The project uses ECMRWF ensemble weather forecasts to drive the streamflow model, and explores both pre- and post-processing of the ensembles for bias correction. Useful results are obtained, and the study has significant potential. I recommend that the paper is accepted pending major revisions.*

We thank the reviewer for the assessment. We appreciate the reviewer's opinion about the potential of the study and the valuable suggestions to improve the manuscript. Below are our responses to the comments and points raised.

*Detailed comments:*

**Comment:** *1. The paper repeatedly refers to HBV as a spatially lumped model. This isn't just terminology, as around lines 20-25 of page 15, the manuscript seems to imply that the model assumes a snowpack to be present (or absent) across the entire model domain. There are a few versions of HBV, but it's normally viewed as semi-distributed, using (at a minimum) elevation bands.*

**Reply:** We appreciate the comment but see little opportunity to soundly add more detail in representing elevation bands. We have chosen to apply a lumped version of the HBV model, without elevation bands, because the available measurement data does not justify to enter multiple bands. For the area only five meteorological stations are available, which cannot be used to represent multiple elevation bands over the complete elevation distribution of the catchment. Following a first analysis on streamflow simulation results, there was no clear signal that model performance is largely affected by lumping so we considered it plausible to rely on the lumped model approach. If requested by the reviewers we could add a comment to Sect. 3.1.1 to address these considerations.

**Comment:** *2. The manuscript makes a good point on lines 29-30 of page 1 about socio-economic development increasing the impacts of extreme hydrometeorological events. It also probably bears mentioning that climate changes, both natural and anthropogenic, may further exacerbate these impacts. See Perkins, Pagano, and Garen, "Innovative operational seasonal water supply forecasting technologies," Journal of Soil and Water Conservation, 2009; and Fleming, "Demand modulation of water scarcity sensitivities to secular climatic variation: theoretical insights from a computational maquette," Hydrological Sciences Journal, 2016.*

**Reply:** We thank the reviewer for the comment and refer to P1 Line 29-30:

"Accurate forecasting becomes increasingly more important because frequency and magnitude of low and high streamflow events are projected to increase in many areas in the world as a result of climate change (IPCC, 2014). Due to socio-economic development  the impacts of extreme events further increase (Bouwer et al., 2010; Fleming, 2016; Rojas et al., 2013; Wheater and Gober, 2015)."

The first sentence aims to mention that climate change exacerbate both low and high streamflow events. Following the reviewer's comment we will add "as a result of climate change" to make the statement more explicit. The paper by Fleming (Hydrological Sciences Journal, 2016) is a good reference for the second sentence.

**Comment:** *3. Terms could stand to a little better defined. For example, most flood and water supply forecasters who I know would regard "short-term" forecasts as having lead times of 0-10 days, and "long-term" forecasts as having lead times of weeks to months. So what the authors refer to here as "medium-term" would be referred to as "short-term" by many if not most others working in the field. And no effort is made here to distinguish medium-*

*term from short-term hydrologic forecasting. More broadly, some of the wording throughout the manuscript would benefit from a re-think for better clarity and precision.*

**Reply:** To be consistent with respect to forecast windows, we explicitly define "medium-range" forecasts and follow the definition for "medium-range" by the World Meteorological Organization, which is also followed by ECMWF (ECMWF, 2012). WMO defines medium-range as forecasts with lead times from 3 days to 10 days, and we also refer to Olsson and Lindström (2008), Renner et al. (2009), and Roulin and Vannitsem (2005). We note that Bennett et al. (2014) refer to this range of lead times as "short-term" forecasts, so there is ambiguity. We opt to keep the term "medium-range" instead of changing it to "short-range", to remain consistent with definitions commonly used in meteorology.

In this paper the term "medium-range" is just used as a generic term to characterize the forecasting system. We do not explicitly distinguish short-range forecasts and medium-range forecasts, because in the analyses there is always referred to specific lead times.

*Comment: 4. Why is only meteorological forecast uncertainty incorporated into the ensemble model? It's commonplace in the research literature for forecast models to include both meteorological uncertainty (NWP ensemble) and hydrologic model parameterization uncertainty (ensemble of hydrologic parameter values). This work is starting to make its way into operational practice too. Providing some justification for this choice might be a good idea.*

**Reply:** We agree to the comment, but argue that only meteorological forecast uncertainty is incorporated because this study aims to identify effects of the ECMWF meteorological forecasts on the quality and skill of streamflow forecasts. Additionally incorporating hydrological model uncertainty, parameter uncertainty and initial condition uncertainty would (partly) obscure this relation. In addition, Bennett et al. (2014), and Cloke and Pappenberger (2009) state that uncertainties in meteorological forecasts are the largest source of uncertainty beyond 2-3 days, and that only uncertainty in meteorological forecasts is incorporated in many studies (Bennett et al., 2014). We will add the above in Sect. 3.1.

*Comment: 5. The description of the model implementation isn't quite adequate. What was the calibration-testing split, and what were the model performances during both phases? And it's stated that the objective function selected for calibration is "Y", which apparently combines the Nash-Sutcliffe efficiency with a volumetric error measure. Objective function selection is a key step in model calibration, and more information needs to be provided, starting with an explicit mathematical definition for "Y".*

**Reply:** We refer to P5 Line 16-23 where the calibration procedure is explained. The equations for Y, NS and $E_{RV}$ are directly accessible in the cited references and we therefore hesitate to add the equations. The calibration and validation performances are listed in Table 4 and referred to on P10 Line 25-28.

*Comment: 6. The updating of initial states was performed here for the slow-runoff and fast-runoff reservoirs. That's interesting and useful, but why was SWE not selected as the object of this data assimilation exercise? It seems like it would be a more rewarding, and certainly more conventional, choice in this northern continental European mountain catchment.*

**Reply:** We thank the reviewer for this thoughtful comment. If the catchment would have exclusively or mainly a snow regime, we would agree that updating of the snow storage would be a more logical choice. However, the catchment does not have an exclusive snow regime, but it has a mixture of regimes (also represented in Figure 4). Moreover, essential to the success of the updating procedure is the availability and quality of data on snow cover, and we consider this investigation to be out of the focus of this paper.

We have used streamflow measurements on the day preceding the forecast issuing day to update the slow and fast runoff reservoirs. This is possible because in the HBV model there is a direct connection between these reservoirs and discharge. Such a direct connection does not exist with the snow storage reservoir. Daily streamflow measurements commonly have a high autocorrelation, so it can be expected that observed streamflow on day *t-1* provides information about the storage in the slow runoff reservoir and fast runoff reservoir on day *t*. We expect that the correlation between snow water equivalent on day *t* and streamflow on day *t-1* will be much lower, and therefore updating of the snow storage using streamflow measurements will be less effective.

*Comment: 7, The literature review of ensemble hydrologic forecasting, pre- and post-processing for bias corrections, and data assimilation and model updating, is a good start but seems a little light. Citing more work would provide valuable context to the paper. A reasonable place to start might be recent work by Dominique Bourdin at the University of British Columbia and Hamid Moradkhani at Portland State University.*

**Reply:** We thank the reviewer for suggesting these sources of additional relevant literature, especially the work by Moradkhani about pre- and post-processing (e.g. Khajehei and Moradkhani, 2017; Madadgar et al., 2014) and updating and data assimilation (e.g. Liu et al., 2012; Pathiraja et al., 2015; Yan and Moradkhani, 2016), and the work by Bourdin which contains recent developments in ensemble streamflow forecasting (Bourdin et al., 2012; Bourdin and Stull, 2013). We will further study the papers by Moradkhani and Bourdin and use this to further extend the context of the paper on ensemble streamflow forecasting, pre- and post-processing and updating procedures.

*Comment: 8. Some of the specific conclusions seem a little surprising. That's great, but it also means they'd benefit from additional discussion. In particular, the paper concludes in section 4.2.1 that the quality of the forecasts at lead times of less than 3 days is dominated by hydrologic initial conditions, and the weather forecasts become the dominant source of predictive skill after that. This would be a reasonable conclusion for a large or flat basin, but for a small, steep mountain river it seems a little surprising – these are typically flashy systems that respond to rain or snowmelt inputs within a day or so. Indeed, a few pages later near the end of section 5, the paper states that "in the hydrological model the lag time between a rainfall event and the streamflow peak is set to 1 day." It also seems that conclusions like this, which attempt to attribute predictive skill (and therefore also predictive error) to various different sources, might be difficult to make convincingly without using a more statically sophisticated and exhaustive data assimilation procedure, incorporating ensembles of hydrologic models and/or model parameters, etc.*

**Reply:** We thank the reviewer for this comment and will further explain the observations of the reviewer. Regarding the comment that "the quality of the forecasts at lead times of less than 3 days is dominated by hydrologic initial conditions, and the weather forecasts become the dominant source of predictive skill after that" is "a little surprising": our results show that this depends on the streamflow category and the streamflow generating process. Short-rain generated high streamflows, snowmelt generated high streamflows and snow accumulation generated low flows are skillfully forecasted by the meteorological forecasts after 1 or 2 days, which could be expected for these fast processes and confirms the expectations of the reviewer. For long-rain generated high streamflows, medium streamflows and precipitration deficit generated low streamflows the maximum skill is observed at larger lead times, because for these processes both the forecasts and the alternative forecasts are dominated by the initial conditions at small lead times.

**References**

Bennett, J. C., Robertson, D. E., Shrestha, D. L., Wang, Q. J., Enever, D., Hapuarachchi, P. and Tuteja, N. K.: A System for Continuous Hydrological Ensemble Forecasting (SCHEF) to lead times of 9 days, J. Hydrol., 519, 2832–2846, doi:10.1016/j.jhydrol.2014.08.010, 2014.

Bourdin, D. R. and Stull, R. B.: Bias-corrected short-range Member-to-Member ensemble forecasts of reservoir inflow, J. Hydrol., 502, 77–88, doi:10.1016/j.jhydrol.2013.08.028, 2013.

Bourdin, D. R., Fleming, S. W. and Stull, R. B.: Streamflow Modelling: A Primer on Applications, Approaches and Challenges, Atmosphere-Ocean, 50(4), 507–536, doi:10.1080/07055900.2012.734276, 2012.

Bouwer, L. M., Bubeck, P. and Aerts, J. C. J. H.: Changes in future flood risk due to climate and development in a Dutch polder area, Glob. Environ. Chang., 20(3), 463–471, doi:10.1016/j.gloenvcha.2010.04.002, 2010.

Cloke, H. L. and Pappenberger, F.: Ensemble flood forecasting: A review, J. Hydrol., 375(3–4), 613–626, doi:10.1016/j.jhydrol.2009.06.005, 2009.

ECMWF: Describing ECMWF's forecasts and forecasting system, edited by B. Riddaway, ECMWF Newsl., 133, 11–13 [online] Available from: http://old.ecmwf.int/publications/newsletters/pdf/133.pdf, 2012.

Fleming, S. W.: Demand modulation of water scarcity sensitivities to secular climatic variation: theoretical insights from a computational maquette, Hydrol. Sci. J., 61(16), 2849–2859, doi:10.1080/02626667.2016.1164316, 2016.

IPCC: Climate Change 2014: Synthesis Report. Contribution of Working Groups I, II and III to the Fifth Assessment Report of the Intergovernmental Panel on Climate Change, edited by Core Writing Team, R. K. Pachauri, and L. A. Meyer, IPCC, Geneva, Zwitzerland. [online] Available from: http://www.ipcc.ch/pdf/assessment-report/ar5/syr/SYR_AR5_FINAL_full.pdf, 2014.

Khajehei, S. and Moradkhani, H.: Towards an improved ensemble precipitation forecast: A probabilistic post-processing approach, J. Hydrol., 546, 476–489, doi:10.1016/j.jhydrol.2017.01.026, 2017.

Liu, Y., Weerts, A. H., Clark, M., Hendricks Franssen, H. J., Kumar, S., Moradkhani, H., Seo, D. J., Schwanenberg, D., Smith, P., Van Dijk, A. I. J. M., Van Velzen, N., He, M., Lee, H., Noh, S. J., Rakovec, O. and Restrepo, P.: Advancing data assimilation in operational hydrologic forecasting: Progresses, challenges, and emerging opportunities, Hydrol. Earth Syst. Sci., 16(10), 3863–3887, doi:10.5194/hess-16-3863-2012, 2012.

Madadgar, S., Moradkhani, H. and Garen, D.: Towards improved post-processing of hydrologic forecast ensembles, Hydrol. Process., 28(1), 104–122, doi:10.1002/hyp.9562, 2014.

Olsson, J. and Lindström, G.: Evaluation and calibration of operational hydrological ensemble forecasts in Sweden, J. Hydrol., 350(1–2), 14–24, doi:10.1016/j.jhydrol.2007.11.010, 2008.

Pathiraja, S., Marshall, L., Sharma, A. and Moradkhani, H.: Hydrologic modeling in dynamic catchments: A data assimilation approach, Water Resour. Res., 52(5), 3350–3372, doi:10.1002/2015WR017192, 2015.

Renner, M., Werner, M. G. F., Rademacher, S. and Sprokkereef, E.: Verification of ensemble flow forecasts for the River Rhine, J. Hydrol., 376(3–4), 463–475, doi:10.1016/j.jhydrol.2009.07.059, 2009.

Rojas, R., Feyen, L. and Watkiss, P.: Climate change and river floods in the European Union: Socioeconomic consequences and the costs and benefits of adaptation, Glob. Environ. Chang., 23(6), 1737–1751, doi:10.1016/j.gloenvcha.2013.08.006, 2013.

Roulin, E. and Vannitsem, S.: Skill of Medium-Range Hydrological Ensemble Predictions, J. Hydrometeorol., 6(5), 729–744, doi:10.1175/JHM436.1, 2005.

Wheater, H. S. and Gober, P.: Water security and the science agenda, Water Resour. Res., 51(7), 5406–5424, doi:10.1002/2015WR016892, 2015.

Yan, H. and Moradkhani, H.: Combined assimilation of streamflow and satellite soil moisture with the particle filter and geostatistical modeling, Adv. Water Resour., 94, 364–378, doi:10.1016/j.advwatres.2016.06.002, 2016.

---

## Author Comment (AC3) · 18 Mar 2017

**Response to Interactive comment Anonymous Referee #3**

*The authors proposed a methodology to give insight in the performance of ensemble streamflow forecasting systems in three streamflow categories (low, medium and high) and related runoff generating processes from lead times of 1 day to 10 day with a case study in a mountainous river catchment of less than 1000 sqr km in Poland. The quantitative precipitation forecasts and temperature forecasts extracted from the European Centre for Medium-Range Weather Forecasts (ECMWF) are averaged with catchment as input of a lumped hydrological (HBV) to generate ensemble streamflow. Several intensively used verification measures (CRPS, CRPSS, Rank histogram, Reliability diagram and ROC) are selected to evaluate the ensemble forecasts. Additionally, the pre-processing, post processing and updating of model initial states are adopted to improve the behavior of the system.*

*Generally speaking, the study gave an interesting investigation on the assessment of hydrological ensemble prediction system on different runoff processes including snowmelt, short-rain flood and so on, and a further analysis was made on the uncertainty source of these varied hydrometeorological conditions. There I suggest accept this manuscript after a moderate revision.*

We thank the reviewer for the assessment. We appreciate the reviewer's opinion about the study and the valuable suggestions to improve the manuscript. Below are our responses to the comments and points raised.

*There are a few issues list below that the authors should address:*

**Comment:** *1) The logic in Paragraph 2 and 3 of Section 1 needs to be perfect. Some irrelevant statements can be removed, eg. SOME CONTENTS from Line 10 to Line 15 in Page 2 about EFAS are unnecessary to some degree.*

**Reply:** We agree with this comment. The text below is a revised version of paragraph 2 and 3 of Sect. 1.

"A number of studies investigated the performance of ensemble forecasting systems for different lead times, e.g.  Alfieri et al. (2014) for the European Flood Awareness System (EFAS), and Bennett et al. (2014), Olsson and Lindström (2008), Renner et al. (2009) and Roulin and Vannitsem (2005) for several catchments varying in size and other characteristics.  These studies all found a deterioration of performance with increasing lead time. ~~EFAS serves to provide high streamflow forecasts in large European river catchments for lead times between 3 and 10 days (Thielen et al., 2009). Relative to hydrological persistency the system skilfully forecasts high streamflow events for all lead times up to 10 days, with increasing skill for larger upstream areas (Alfieri et al., 2014). In EFAS critical flood warning thresholds are based on simulated streamflow, because model results and streamflow measurements can largely deviate (Thielen et al., 2009). EFAS is aimed at providing early warnings of possible flooding, instead of providing specific river streamflow forecasts (Demeritt et al., 2013)., in contrast totoP~~revious studies did not assess effects of runoff processes, like snowmelt and extreme

rainfall events, on the performance of the ensemble forecasts. The only study we found that touches on this is the study by Roulin and Vannitsem (2005), who described that their high streamflow forecasting system is more skilful for the winter period than for the summer period. For two Belgium catchments the high streamflow forecasting system of Roulin and Vannitsem (2005) is more skilful for the winter period than the summer period. Previous studies did not assess effects of runoff processes, like snowmelt and extreme rainfall events, on the performance of the ensemble forecasts.

Next to an assessment of performance of forecasts, iInformation on the relative importance of uncertainty sources in forecasts is helpful essential to improve the forecasts effectively (Yossef et al., 2013). A number of studies report on how errors in the meteorological forecasts and the hydrological model contribute to errors in medium-range hydrological forecasts. Demargne et al. (2010) show that hydrological model uncertainties (initial conditions, model parameters and model structure) are most significant at short lead times. However, tThis also depends on the streamflow category . : hHydrological model uncertainties significantly degrade the evaluation score up to a lead time of 7 days for all flows and up to a lead time of 2 days for the very high streamflow events. Renner et al. (2009) found an underprediction of low forecast probabilities (few ensemble members over a high streamflow threshold), which they attribute to the meteorological forecasts having (insufficient variability). On the other hand, the high forecast probabilities (low threshold) are overpredicted, which Renner et al. (2009) attribute to both the hydrological model and the meteorological input data. Olsson and Lindström (2008) found an underestimation of the spreadunderdispersion of ensemble flood forecasts, to an extent that decreases with lead time. They conclude that the meteorological forecasts and the hydrological model have a comparable contribution to this underestimation. In addition, Olsson and Lindström (2008) show overprediction of forecast probabilities over high thresholds, which they mainly primarily attribute to the meteorological forecasts. Regarding low streamflow forecasts, Demirel et al. (2013) concluded that uncertainty of hydrological model parameters has the largest effect, whereas and meteorological input uncertainty has the smallest effect on low streamflow forecasts. Based on those studies we can say that for high streamflow forecasts uncertainties in the meteorological forecasts are dominant, whereas for low streamflow forecasts the uncertainties in the hydrological model become more important."

*Comment: 2) Lines18-20 Page 6: A further explanation is expected why the training period is defined from 2011-2013 while the years previous to 2011 is used to validation.*

**Reply:** Our approach was triggered by practical considerations. We have serious doubts about the quality of the observation data in 2007: for the hydrological year 2007 (1 Nov 2006 – 31 Oct 2007) the agreement between observed discharge and simulated discharge with observed precipitation and temperature is poor (see table below). Therefore the hydrological year 2007 was excluded from further analysis.

The performance of the hydrological model for the hydrological year 2008 also raised some doubts about the quality of the observation data during this year. For this reason we started the pre- and post-processing with 2012-2013 (just two hydrological years to have a sufficiently long evaluation period left) as the training period, and we validated the pre- and post-processing procedures on both 2008-2011 and 2009-2011. There was no significant difference in validation performance of the pre- and post-processing procedures between these two periods and also the hydrographs of observations and simulations do not indicate poor quality of observation data for 2008, so in the end we included 2008 in the validation period.

*Table 1: Validation performance per hydrological year*

| Hydrological year | NS | $E_{RV}$ [%] | Y |
|---|---|---|---|
| 2007 | -1.34 | 43.41 | -0.94 |
| 2008 | 0.22 | 17.14 | 0.19 |
| 2009 | 0.53 | -4.67 | 0.51 |
| 2010 | 0.93 | 0.07 | 0.93 |
| 2011 | 0.59 | 6.20 | 0.55 |
| 2012 | 0.62 | 19.47 | 0.52 |
| 2013 | 0.46 | 12.79 | 0.41 |

We noticed that the validation performance numbers in Table 4 of the paper do include the hydrological year 2007. We will recalculate these numbers after excluding 2007.

*Comment: 3) In Section 3.2, it is not necessary to introduce all the evaluation scores in details, for the CRPS, CRPSS, Reliability diagram and ROC can be regarded as "industry standards" in ensemble forecasting, so simply citing the relevant references.*

**Reply:** We agree to the comment and will omit general information about the evaluation scores (P7 Line 13-15, Line 18-20, P8 Line 14-21, P9 Line 2-5, Line 12-15, Line 16-18). In Sect. 3.2 we will address what aspect of forecast quality a score evaluates and refer to other studies for further details.

*Comment: 4) In Section 4.1.2, it is confusing that since the QM pre-processing brings improvement to the precipitation and temperature forecasts, why the conclusion is that the strategy 0 results in the best CRPS.*

**Reply:** We agree to the reviewer that this is a remarkable result. The results indicate that the slight improvement of the meteorological forecasts by the pre-processing procedure loses its effect after propagating through the hydrological model. We will add this finding to the conclusion of the paper (P17 Line 12).

*Comment: 5) The figures about rank histograms and reliability diagrams are missing or not shown intentionally?*

**Reply:** The figures about rank histograms, reliability diagrams and ROC curves are not shown by intention to keep the paper short. However, we think that these results are relevant and therefore we described them in words. We agree that this makes reading of the paper difficult and the results nontransparent. We could make the figures available by a supplement to the paper.

*Comment: 6) The catchment area is less than 1000km2 and the data used are daily. For flood forecasting in such catchment area, is it daily data too coarse? Perhaps 3h or 6h subdaily data are more useful for flood forecasting in such area. Please make it an elaborate story.*

**Reply:** We thank the reviewer for the comment but note that discharge measurements are available at a daily resolution. For this reason we applied and evaluated the forecasting system at a daily time step. When focusing on short-range forecasts (lead times of 0-2 days), we agree that smaller time steps are preferred for a mountainous catchment of about 1000 km$^2$ like the Biala Tarnowska catchment. We focus on medium-range forecasts (0-10 days), for which the very quick streamflow response is less important.

*Comment: 7) For flood forecasting, flood peak, volume and peak time are all important. Can these be analyzed in the study?*

**Reply:** We agree with the reviewer that, in addition to discharge, the peak streamflow, volume and peak time are important, particularly for operational high streamflow forecasting systems. Despite

the relevance, we propose not to include the analyses of these aspects in the paper. Looking at the number of pages the paper already has we must be selective in what we can include. Moreover, essential to the topic of the paper is that next to high streamflows we also evaluate the streamflow forecasting system on low streamflows and medium streamflows. In view of the paper length already we cannot evaluate low streamflows, medium streamflows and high streamflows on all relevant aspects, such as duration and discharge deficits regarding low streamflows.

**Comment:** *8) Page 9: It is not very clear how the errors are contributed in Section 3.3. Why can CRPSsim/CRPSmeans represent the error contribution? Please add more details.*

**Reply:** Evaluation against observed discharge ($CRPS_{meas}$) is affected by errors from the meteorological forecasts, the hydrological model and measurement errors. By evaluation against simulated discharge based on observed precipitation and temperature ($CRPS_{sim}$), the ensemble streamflow forecasts and the reference streamflow contain similar hydrological model errors and no streamflow measurement errors, so these are eliminated. If we neglect measurement errors we get:

$$\frac{CRPS_{sim}}{CRPS_{meas}} \sim \frac{meteorological\ forecast\ errors}{meteorological\ forecast\ errors + hydrological\ model\ errors}$$

If this ratio is low the hydrological model errors are dominant and if this ratio is high the meteorological forecast errors are dominant. The same approach is used by Demargne et al. (2010), Olsson and Lindström (2008) and Renner et al. (2009).

To clarify this explanation we will add the equation above.

**References**

Alfieri, L., Pappenberger, F., Wetterhall, F., Haiden, T., Richardson, D. and Salamon, P.: Evaluation of ensemble streamflow predictions in Europe, J. Hydrol., 517, 913–922, doi:10.1016/j.jhydrol.2014.06.035, 2014.

Bennett, J. C., Robertson, D. E., Shrestha, D. L., Wang, Q. J., Enever, D., Hapuarachchi, P. and Tuteja, N. K.: A System for Continuous Hydrological Ensemble Forecasting (SCHEF) to lead times of 9 days, J. Hydrol., 519, 2832–2846, doi:10.1016/j.jhydrol.2014.08.010, 2014.

Bürger, G., Reusser, D. and Kneis, D.: Early flood warnings from empirical (expanded) downscaling of the full ECMWF Ensemble Prediction System, Water Resour. Res., 45(W10443), doi:10.1029/2009WR007779, 2009.

Demargne, J., Brown, J., Liu, Y., Seo, D. J., Wu, L., Toth, Z. and Zhu, Y.: Diagnostic verification of hydrometeorological and hydrologic ensembles, Atmos. Sci. Lett., 11(2), 114–122, doi:10.1002/asl.261, 2010.

Demirel, M. C., Booij, M. J. and Hoekstra, A. Y.: Effect of different uncertainty sources on the skill of 10 day ensemble low flow forecasts for two hydrological models, Water Resour. Res., 49(7), 4035–4053, doi:10.1002/wrcr.20294, 2013.

Fundel, F., Jörg-Hess, S. and Zappa, M.: Monthly hydrometeorological ensemble prediction of streamflow droughts and corresponding drought indices, Hydrol. Earth Syst. Sci., 17(1), 395–407, doi:10.5194/hess-17-395-2013, 2013.

Komma, J., Reszler, C., Blöschl, G. and Haiden, T.: Ensemble prediction of floods - catchment non-linearity and forecast probabilities, Nat. Hazards Earth Syst. Sci., 7(4), 431–444, doi:10.5194/nhess-7-431-2007, 2007.

Olsson, J. and Lindström, G.: Evaluation and calibration of operational hydrological ensemble forecasts in Sweden, J. Hydrol., 350(1–2), 14–24, doi:10.1016/j.jhydrol.2007.11.010, 2008.

Renner, M., Werner, M. G. F., Rademacher, S. and Sprokkereef, E.: Verification of ensemble flow forecasts for the River Rhine, J. Hydrol., 376(3–4), 463–475, doi:10.1016/j.jhydrol.2009.07.059, 2009.

Roulin, E. and Vannitsem, S.: Skill of Medium-Range Hydrological Ensemble Predictions, J. Hydrometeorol., 6(5), 729–744, doi:10.1175/JHM436.1, 2005.

Thielen, J., Bartholmes, J., Ramos, M. H. and De Roo, A.: The European Flood Alert System - Part 1: Concept and development, Hydrol. Earth Syst. Sci., 13(2), 125–140, doi:10.5194/hess-13-125-2009, 2009.

Verkade, J. S., Brown, J. D., Reggiani, P. and Weerts, A. H.: Post-processing ECMWF precipitation and temperature ensemble reforecasts for operational hydrologic forecasting at various spatial scales, J. Hydrol., 501, 73–91, doi:10.1016/j.jhydrol.2013.07.039, 2013.

Yossef, N. C., Winsemius, H., Weerts, A., Van Beek, R. and Bierkens, M. F. P.: Skill of a global seasonal streamflow forecasting system, relative roles of initial conditions and meteorological forcing, Water Resour. Res., 49(8), 4687–4699, doi:10.1002/wrcr.20350, 2013.

Zappa, M., Jaun, S., Germann, U., Walser, A. and Fundel, F.: Superposition of three sources of uncertainties in operational flood forecasting chains, Atmos. Res., 100(2–3), 246–262, doi:10.1016/j.atmosres.2010.12.005, 2011.

---

## Author Response (AR1)

**Author response HESS-2016-584**

**8 May 2017**

Title: Performance of ensemble streamflow forecasts under varied hydrometeorological conditions

Authors: Harm-Jan F. Benninga, Martijn J. Booij, Renata J. Romanowicz, Tom. H.M. Rientjes

Dear editor,

Thank you for your consideration of the paper. Based on the comments by the three reviewers on the first version of the manuscript we have revised the manuscript regarding explanation of methods, explanation of results and English writing. We have added Fig. 5 and additional background figures in a supplement. The revised manuscript is uploaded.

This document contains a point-by-point reply to the comments and a marked-up manuscript. Page and line numbers refer to the first version of the manuscript. We have updated revised texts in the point-by-point reply with the text in the uploaded revised manuscript (minor changes compared to the responses on 18 March 2017). In the cases that our response to a comment has changed compared to 18 March 2017, we have indicated this by keeping the original reply (18 March 2017) and adding an additional reply below it (8 May 2017).

**Response to Interactive comment Anonymous Referee #1**

**General comments**

**Comment:** This manuscript presents an interesting analyse of the performance of hydrological ensemble predictions. The skills are screened according the regime (low and high streamflow) and the generating processes (snow melt, short rain, long rain floods etc.). This study further disentangles hydrological model errors and errors from meteorological forcing. The methodology is applied to a mountainous catchment. The combination of existing methodologies is pertinent and is worth being published in HESS.

However the reading is not easy and a major revision is necessary. Some information is redundant in the introduction, methodology and results sections and long lists of references are not always necessary. The focus should be made on the main contribution of the paper i.e. the analysis of the skill for different hydrometeorological conditions and skip or shorten secondary experiments. Some validation methodologies are described but their results are not shown. A balance should be found: either shorten the description or include those results. Some suggestions are given in the specific comments. The English should be improved.

**Reply:** We thank the reviewer for the assessment. We appreciate the reviewer's opinion about the study and the valuable suggestions provided to improve the manuscript. Below are our responses to the comments and points raised.

The reviewer's suggestion to improve the flow of the paper is valuable, and the specific comments contain many relevant points for this.

With respect to the comment to increase the focus of the paper on the main scientific innovation, we will leave out the additional updating experiment, which has also not been used because it was unsuccessful (P5 Line 31 - P6 Line 2, P11 Line 3 - P11 Line 5).

Regarding the experiments on pre- and post-processing of the ensemble forecasts we consider this important and propose not to remove it from the paper. The procedure is common, so removing it will presumably result in doubts about why we have not applied a correction procedure. The results of this experiment are quite striking and we will add a figure with CRPS values for the different preand post-processing strategies showing this finding (see Figure 1).

Further replies to this comment follow below in response to the specific comments.

Figure 1: CRPS of the post-processing strategies over the validation period 2008-2011

**Comment:** The authors are using ensemble predictions from ECMWF from 2007 to 2013 with a training of the pre- and post-processing during two water years between 2011 and 2013. They associate the failure of the quantile mapping for post-processing method to the short time series of forecasts for training and to the inconsistency of the bias between the training and the validation period. They forget that the ensemble prediction system has undergone many changes during this period including spatial resolution changes. This is why retrospective forecasts are available since long and provide samples of 18 to 20 years back for post-processing purposes. Re-forecasts have been widely used and reported in the literature. These meteorological re-forecasts have also been used for the preparation of hydrological re-forecasts for the statistical postprocessing of hydrological ensemble predictions.

**Reply:** It is correct that we used meteorological forecasts from a system that has undergone changes. The TIGGE data portal contains the operational forecasts from meteorological forecast centres. We agree that this affects the pre-processing and post-processing results and we thank the reviewer for this suggestion. We will add a statement to Page 15 Line 15-16 that the joint distribution of measurements and forecasts is nonhomogeneous in time, because the meteorological forecast system has undergone changes during our analysis period (Mladek, 2016):

https://software.ecmwf.int/wiki/display/TIGGE/Model+upgrades#Modelupgrades-ECMWF

**Comment:** Figure 5 to 9 are the core of the paper. They will gain value if the plots are associated with confidence intervals.

**Reply:** CRPS and CRPSS are the main evaluation scores that we used. In recent literature these scores are commonly applied without associated confidence intervals or statistical tests, by Demargne et al. (2010), Hersbach (2000), Pappenberger et al. (2015), Renner et al. (2009), Verkade et al. (2013), and Ye et al. (2014). We agree to the suggestion that confidence intervals around the CRPS values would add value to the figures, but we consider establishing such confidence intervals outside the focus of this paper.

**Comment:** The use of the term "perfect forecast" is questioned because it is neither a forecast nor perfect and, would the future meteorological forcing be known, predictions with the model would include growing errors due to initial conditions as somehow shown in Table 5.

**Reply:** We appreciate the comment. The term "perfect forecast" was introduced by Olsson and Lindström (2008), but the term is somewhat misleading. For the same concept, Renner et al. (2009) used the term "baseline simulation", Demargne et al. (2010) used the term "simulated flow", Verkade et al. (2013) used the term "simulated streamflow" and Bennett et al. (2014) used the term "perfect-rainfall-forced forecasts". We propose to use "observed meteorological input forecasts".

**Specific comments**

Comment: P1, L20-24 Should be rephrased e.g. too many occurrence of "improve".

**Reply:** We agree to the comment and will change it to:

"To improve the performance of the forecasting system for high streamflow events, in particular the meteorological forecasts require improvementare crucial. For low streamflow forecasts, It is recommended to calibrate the hydrological model specifically on low streamflow conditions and high streamflow conditions. the hydrological model should be improved. The study It is further recommendeds improving that the reliability dispersion (reliability) of the ensemble streamflow forecasts is enlarged by including the uncertainties in hydrological model parameters and initial conditions, and by improving enlarging the dispersion of the meteorological input forecasts."

**Comment:** P3 L23-P4, L3 How do you correct measurement? Do you correct each station for the difference between the elevation of the station and the average of the elevation in the area defined by the intersection of the Thiessen polygon corresponding to the station and the watershed? Then average the corrected values of the stations using their relative contribution to the catchment area as weights?

**Reply:** The assumption of the reviewer is correct: this is the procedure that we used. We will revise the text to make this clear:

"Precipitation and, temperature and streamflow measurement series are available from five meteorological stations and streamflow measurement series are available from one discharge gauging station, at a daily time interval for the period 1 January 1971 to 31 October 2013, and provided by the Polish Institute of Meteorology and Water Management. Precipitation and temperature data from 5 measurement stations (Fig. 1) have been selected because of their distribution over the catchment and data series completeness. The data are spatially interpolated based on Thiessen polygons (Fig. 1) to represent catchment averages. Given that meteorological stations are mostly located in valleys and precipitation and temperature vary with elevation, the catchment averages are-may be biased (Panagoulia, 1995; Sevruk, 1997). Following Akhtar et al. (2009), precipitation measurements are corrected using relative correction factors (in %), whereas temperature measurements are corrected using absolute correction factors (in °C). The precipitation correction factorgradient differs considerably between months. For December-February the mean precipitation gradient is 10.5 % 100 m-1, while for March–November the mean precipitation gradient is 5.4 % 100 m-1. Although the number of stations is limited small to accurately determine precipitation and temperature gradients, the calculated precipitation gradients are used because of the clear difference between the two periods. The temperature gradient does not vary much over the year and therefore the global standard temperature lapse rate of 0.65 °C 100 m-1 is applied. The measurements from each station are corrected for the difference between the elevation of the station and the mean elevation its respective Thiessen polygon. To represent catchment averages, the corrected measurements are weighted based on the relative coverage of their Thiessen polygon (Fig. 1). By the corrections the annual mean precipitation increases from 741.2 mm to 768.4 mm and the annual mean potential evapotranspiration decreases from 695.3 mm to 674.4 mm."

*Comment:* P5, L20-21 Equations would be appropriate here in order to define Y, NS and E\_RV.

**Reply:** We hesitate to add the equations since Y, NS and  $E_{RV}$  are defined in the given references.

*Comment:* P5, L28 preceding the first forecast day.

**Reply:** We agree. We will change it to: "the day preceding the forecast issuing day" (from comment on P11, L21).

**Comment:** P5, L32-P6, L2 I would suggest to skip this experiment or, if impossible to skip, tell already that it failed (according to P11, L3-5). This is to lighten the methodologies to keep in mind until the result section.

**Reply:** We agree to leave this out. Also see the response to the first general comment.

*Comment:* P6, L31-P7, L11 Some information (and references) is redundant with the sub-sections.

**Reply:** We agree. Also looking at comment 3 by Reviewer 3 we will omit general information about the evaluation scores, but focusing on what aspect on forecast quality each score evaluates and citing the relevant references.

Comment: P7, L1 Three properties of probabilistic forecast quality ....

**Reply:** We do not understand this comment.

Comment: P7, L8 "The histograms accompanying ..." the histograms of what?

Reply: We will change this sentence to:

"The histograms accompanying reliability diagrams are used to evaluate sharpness. To evaluate sharpness, we employ the histograms that show the sample size of the forecast probability bins used to establish the reliability diagrams (Ranjan, 2009; Renner et al., 2009; WMO, 2015)."

**Comment:** P7, L20-21 "CRPS approaches the average value of the evaluated variable" What do you mean with "approaches"?

Reply: We will change "approaches" to "converges to".

**Comment:** P7, L24-27 "and compares the forecasts with a relevant alternative forecast" somehow redundant with the beginning of the sentence.

**Reply:** We will change this sentence to:

"Normalizing the CRPS against the CRPS of alternative forecasts eliminates the effect of the magnitude of the investigated variable and compares the forecasts with a relevant alternative forecast (i.e. skill), used by e.g. Bennett et al. (2014), Demargne et al. (2010), Renner et al. (2009), Velázquez et al. (2010) and Verkade et al. (2013).

To eliminate the magnitude of the investigated variable we normalize the CRPS against the CRPS of a relevant alternative forecast, a principle which is also used by Bennett et al. (2014), Demargne et al. (2010), Renner et al. (2009), Velázquez et al. (2010) and Verkade et al. (2013) to evaluate forecast skill."

**Comment:** P8, L1-2 "... argue that this" choice ... these two lines should be rephrased. I would prefer a positive phrasing saying that the choice of another alternative forecast may result in a more robust estimation of forecast skill.

**Reply:** We propose to delete P7 Line 30 – P8 Line 2, because it is not really relevant to explain the procedure that we followed. It explains why we have not applied hydrological persistency or hydrological climatology as alternative forecast set, but we can focus the text on what we have done: using the forecast set with the lowest CRPS values as alternative forecast set, because this set is most difficult to beat in performance.

**Comment:** P8, L22-23 Either provide an equation for the "numerical indicator delta" if it adds to the understanding of the adopted methodology or skip any reference to delta.

Reply: We will remove the reference to delta.

*Comment:* P8, L30-31 "... contain a random element ..." explain how it works for the flatness coefficient.

**Reply:** We will add further explanation about the random element:

"In this case, a random rank is assigned to the measurement from the pool of ensemble members and the measurement that have the same value."

**Comment:** P9, L3 "... for a certain event ..." It would be useful to define "event" and refer to sub-section 3.4 or Table 1.

**Reply:** We will specify "certain event" as "for low streamflow events and high streamflow events (defined in Sect. 3.4)".

*Comment:* P9, L24-28 Almost the same thing is repeated.

Reply 18 March 2017: We will delete P9 Line 24-26.

**Reply 8 May 2017:** On second thought we consider P9 Line 24-26 valuable in the text. The first sentence explains that the streamflow measurement error and the hydrological model error are eliminated by evaluation against observed meteorological input forecasts, whereas the second sentence explains how this is used to investigate the contribution of error sources.

**Comment:** P9, L29 At a first reading, it was tempting to replace this ratio with a CRPSS of sim against meas but the purpose is different and since it is a major tool in this paper, this paragraph should be written with much care.

Reply: We will add the equation below (see also comment 8 by Reviewer 3):

 $\frac{CRPS_{sim}}{CRPS_{meas}} \sim \frac{meteorological forecast errors}{meteorological forecast errors+hydrological model errors}$

If this ratio is low, the hydrological model errors are dominant and if this ratio is high, the meteorological forecast errors are dominant.

**Comment:** P10, L11-12 Are the rules given also by Merz and Blöschl or defined for this catchment based for instance on data from both simulation and observations during the training period?

**Reply:** The study by Merz and Blöschl (2003) is used to characterize the high streamflow generating processes in Table 2. The rules for classification are defined specifically for the study catchment and are based on observations and model simulations. We will change the text to:

"Various runoff contributing generating processes can result in high flows. Table 2 defines the processes and classification rules for classification we use in this study, based on the processes Merz

and Blöschl (2003) distinguish. The rules for classification are based on rainfall observations and snowpack model simulations; at one day before the event because of the time step used in the HBV model."

**Comment:** P10, L16 Do you mean that the distribution of the generating processes shown in the figure is like we can expect for this region?

**Reply:** The reviewer's interpretation is correct. We will change this to:

"Figure 4a presents the distribution of high streamflow generating processes over the year following the classification rules for classification in Table 2. The figure shows an expected distribution of processes for this region. The distribution of processes is typical for this region."

Likewise we will change P10 Line 20-21.

Comment: P10, L19, Table 3 What is the rule for precipitation deficit?

**Reply:** The rule used for classifying an event as a precipitation deficit generated low streamflow is that if there is a low streamflow event and if there is no snowpack present (based on model simulations) we assume that the low streamflow event is caused by a precipitation deficit. We think that the definition in Table 3 is clear.

*Comment:* P11, L21 "preceding day" the day before the forecast issuing day.

**Reply:** We agree and we will change "preceding day" to "day preceding the forecast issuing day" accordingly in the paper (also see comment P5 Line 28).

**Comment:** P11, L28-29 "not shown in the paper" therefore, going back to section 3.1.3, the methodology description should be simpler and not encumber with strategy numbers.

Reply: This comment is discussed in the response to the first comment.

Comment: P10 L20 What do you mean by "reliable distribution"?

Reply: See response to comment P10, L16.

Comment: P12, L13 with more skill instead of "skilful"

**Reply:** Skill is defined as the performance of the streamflow forecast relative to the performance of alternative forecasts. Here we do not mean 'with more skill', but skilful relative to the alternative forecasts.

Comment: P12, L16 "functional" what do you mean?

Reply: We will change "functional" to "plausible".

Comment: P12, L28 "... are in general less predictable by historical measurements ..." please re-phrase

**Reply:** P12 Line 28-29 is partly a repetition of the preceding sentence, so this sentence will be deleted. We will change P12 Line 27-29 to:

"In addition, these events are high streamflow events will be less well captured in by historical measurements, and thus in the alternative forecasts will have lower quality for these events. This is because high streamflow periods are in general less predictable by historical measurements, in particular in small catchments."

**Comment:** P12, L32 "not shown" a figure is missing with the rank histograms for the low streamflow forecasts and for the high streamflow forecasts, two lead times. Apparently, for high flow, the rank histogram is not exactly U-shaped but skewed according to P13, L12-13.

**Reply:** To keep the paper short we chose not to include these figures in the paper. However, we think that the results are relevant and therefore we described them in words. We agree that this makes reading of the paper difficult and the results nontransparent. We could make the figures available by a supplement to the paper.

**Comment:** P13 L10-13 Difficult to figure out ... Please add a figure with the reliability diagrams and corresponding sharpness histograms for the low streamflow forecasts and for the high streamflow forecasts two lead times.

**Reply:** See response to comment P12 L32.

**Comment:** P13 L15-17 Note that good sharpness without reliability is useless.

**Reply:** We agree. We will emphasize this in the conclusion (bullet 1).

Comment: P13, L18 reference already given, please re-phrase.

Reply 18 March 2017: We agree. We will change this to:

"All AUC values are above 0.85, whereas Buizza et al. (1999) consider 0.8 as indicative for good prediction systems which indicates a good resolution of the streamflow forecast system."

**Reply 8 May 2017:** The explanation of evaluation scores in Sect. 3.2 is shortened and the reference is omitted there. Therefore we keep the reference in this section.

Comment: P14, L11-13 "... the below zero skill ... do not result in positive skill ..."

**Reply 18 March 2017:** We agree that this sentence is not well written. We will change the sentence to:

"The below 0 skill of long-rain and snowmelt flood forecasts indicate that the meteorological forecasts at small lead times do not result in positive skill as compared to forecasts based on historical meteorological measurements.

For long-rain floods and snowmelt floods, the meteorological forecasts at small lead times do not result in positive skill as compared to forecasts based on historical meteorological measurements."

**Reply 8 May 2017:** We remove this sentence, because at this point in the paper it is clear that skill is generated by the ECMWF meteorological forecasts compared to historical meteorological measurements.

Comment: P14, L23 What is the amount of this fake drizzle?

**Reply:** This is an interesting question, but we consider this to be out of the focus of this paper.

**Comment:** P14, L24-26 Re-phrase: "... meteorological forecasts accumulated in the forecasting system are better model inputs ..."

Reply: We agree that this sentence is not well written. We will change the sentence to:

"The skill increases for larger lead times, so for larger lead times ECMWF meteorological forecasts accumulated in the forecasting system <del>are better model inputsgive better predictions</del> than historical meteorological measurements for larger lead times." **Comment:** P15, L8 & Figure 10 I would skip this figure which highlights the weakness of drawing such a detailed profile with just a water-year data. The legend is missing for the thin plain lines.

**Reply:** We hesitate to skip this figure, because it illustrates why the pre- and post-processing procedures are not working: the training period and validation period show different bias distributions, because of the short time series.

The thin plain lines are showed in the legend as "Single years 2007-2013". We will add an explanation to the caption that each thin line refers to a single year between 2007 and 2013.

**Comment:** P16, L8-10 Do you have evidence that such coincidence occurs and is the main explanation for the high ratio for short-rain floods?

Reply 18 March 2017: This is an interesting question and we will investigate how often this occurs.

**Reply 8 May 2017:** We have further investigated this question. There are two possible cases if the ensemble forecast is closer to the measured streamflow than to the observed meteorological input forecasts: 1. the observed meteorological input forecast is closer to the measured streamflow (example in Fig. 1), and 2. the ensemble forecast is closer to the measured streamflow (example in Fig. 2). The second case indicates a hydrological model deficiency: in the rainfall-runoff relation or in the flood peak timing. Table 1 lists the numbers associated with both cases, based on CRPS calculations for each day classified as high streamflow.

---

## Author Response (AR2)

**Title:** Performance of ensemble streamflow forecasts under varied hydrometeorological conditions

**Authors:** Harm-Jan F. Benninga, Martijn J. Booij, Renata J. Romanowicz, Tom. H.M. Rientjes

Dear editor,

Thank you for your consideration of the paper. Based on the comments by two reviewers on the second version of the manuscript (2017-05-08) we have revised the manuscript regarding explanation of results and English writing. The revised manuscript has been uploaded.

This document contains a point-by-point reply to the comments and the marked-up revised manuscript. Page and line numbers refer to the unrevised second version of the manuscript (2017-05-08).

**Response to comments Anonymous Referee #1 on manuscript 2017-05-08**

We thank the reviewer for the second assessment. Below are our responses to the comments raised. We have revised the manuscript accordingly.

*General comment: The revised manuscript has improved. This interesting study is worth being published in HESS subject to minor changes.*

*The authors have replied to my comments satisfactorily except about the shortness of the time series used for training the QM. They only retain the possible effect of changes in the ECMWF forecast system. My message advocated also for the use of retrospective forecasts (reforecasts): ECMWF provides such forecasts of past situations using the current operational system since 2008! (See Hagedorn, 2008). Nowadays, they issue reforecasts twice a week, twenty years back, with 11-member ensembles. These reforecasts can be used for post-processing precipitation predictions (or pre-processing in the present context, see numerous literature) and even to prepare hydrological reforecasts for calibrating the post-processing of ensemble streamflow forecast (also some literature). Even if the reforecasts are not included in the TIGGE database and are not used in this study, it is a possible way to improve the methodology. Please reflect this comment (or refute) in the second paragraph of the Discussion section and in the fourth bullet in the Conclusions.*

**Reply:** We thank the reviewer for the suggestion to use the retrospective forecasts that are provided by ECMWF and agree that this is a possible way to improve the effectiveness of the pre- and post-processing procedures. We chose to mention this as a possible way to improve the methodology in the conclusion (see below) and not in the discussion. The discussion reflects the applied research methodology and the conclusion contains possible points to improve the methodology and points for further research.

Conclusion P16 Line 28-29: "A longer time series of forecasts  would  promote the success of pre- and post-processing. ECMWF provides a homogeneous retrospective forecast set, consisting of twice-weekly forecasts with one control and 10 ensemble members over a period of 20 years, that is generated by the current operational system (Hagedorn, 2008; Vannitsem and Hagedorn, 2011; Vitart, 2017)."

*Specific comment: P15, lines 13-16. I suggest to rephrase or even split this sentence in two because it appears first to refer twice to the same thing with different percentages.*

**Reply:** We agree to the comment and have changed this part of text to:

Discussion P15 Line 11-24: "The ratio between the CRPS against observed meteorological input forecasts and the CRPS against streamflow measurements is above 100% for high streamflows, and short-rain floods in particular (Fig. 9b). This means that  forecasts are closer to the measurements than to the observed meteorological input forecasts.  On 28% of the high streamflow days at a lead time of 1 day to 48% of the high streamflow days at a lead time of 10 days, the ensemble forecasts are closer to the measurements than to the observed meteorological input forecasts. On 50% to 66% of these days, the ensemble forecasts are closer to the measurements than the observed meteorological input forecasts are. This indicates a hydrological model deficiency in high streamflow conditions, either from simulating the rainfall-runoff relation or the flood peak timing. The precipitation peak in the measurements and the precipitation peak in the meteorological forecasts  may be shifted one day with respect to each other and this may cause that the timing of the peak of the streamflow forecasts better corresponds to the streamflow measurements. Of the 97 separate peak streamflow days, on 6 days (lead time of 6 days) to 17 days (lead time of 1 day) the flood peak day of the observed meteorological input forecasts does

not match to the peak day of the measurements,  while the peak day of the mean of the ensemble forecasts does match to the peak day of the measurements. This illustrates that the hydrological model deficiency regarding flood peak timing  has a considerable effect on the observed meteorological input forecasts and the ensemble forecasts."

**Comment:** *This revised version of manuscript can be accepted after checking English.*

**Reply:** We have done a thorough new check on English. We refer to the changes in the marked-up manuscript.

**References**

[revised manuscript text omitted]